# WISA: World Simulator Assistant for Physics-Aware Text-to-Video Generation

**Jing Wang**[1,2,§,*]**, Ao Ma**[2,†,*]**, Ke Cao**[2,*]**, Jun Zheng**[1]**, Jiasong Feng**[2]**,**
**Zhanjie Zhang**[2]**, Wanyuan Pang**[3]**, Xiaodan Liang**[1,4,5,‡]

[1]Shenzhen Campus of Sun Yat-Sen University, [2]360 AI Research,
[3]University of Science and Technology Beijing, [4]Peng Cheng Laboratory,
[5]Guangdong Key Laboratory of Big Data Analysis and Processing,
`wangj977@mail2.sysu.edu.cn`

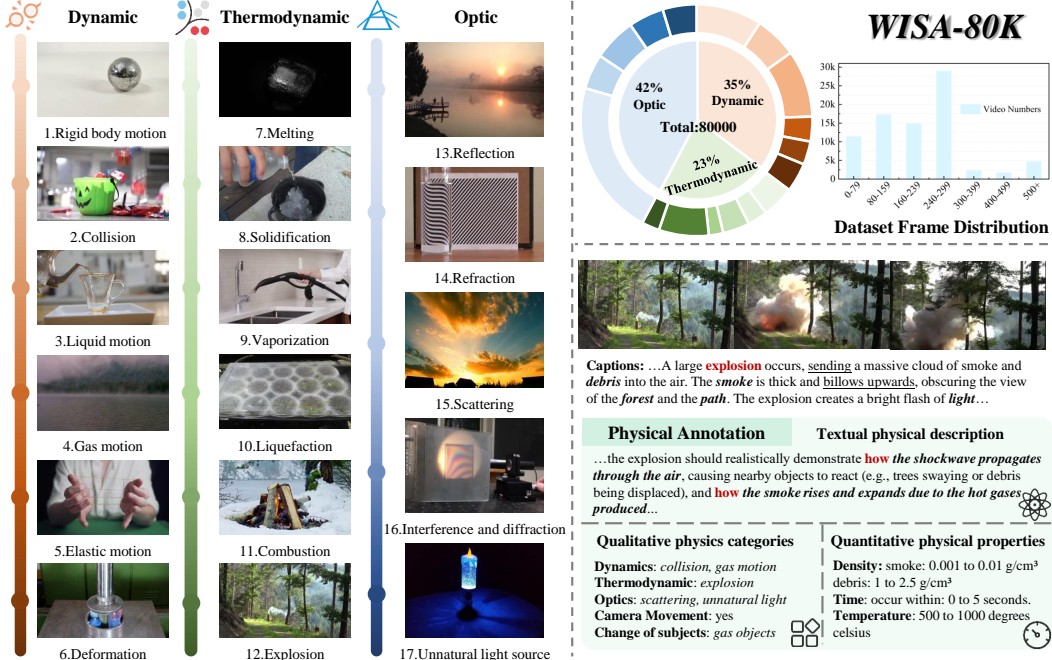

Figure 1: **Overview of our physical dataset WISA-80K**. (Left) Examples of 17 physical phenomena across three physics categories in WISA-80K. (Top right) WISA-80K consists of approximately 80,000 video clips, with 35% related to *Dynamics*, 23% to *Thermodynamics*, and 42% to *Optics*. (Top right) Distribution of frame counts across all videos in WISA-80K. (Bottom right) An example of physical annotation in WISA-80K.

## Abstract

Recent advances in text-to-video (T2V) generation, exemplified by models such as Sora and Kling, have demonstrated strong potential for constructing world simulators. However, existing T2V models still struggle to understand abstract physical principles and to generate videos that faithfully obey physical laws. This limitation stems primarily from the lack of explicit physical guidance, caused by a significant gap between high-level physical concepts and the generative capabilities of current models. To address this challenge, we propose the **W**orld **S**imulator **A**ssistant (**WISA**), a novel framework designed to systematically decompose and integrate physical principles into T2V models. Specifically, WISA decomposes physical knowledge into three hierarchical levels: textual physical descriptions, qualitative physical categories, and quantitative physical properties. It then incor-

---

[*]Equal Contribution. [‡]Corresponding Authors. [†]Project Leader. [§]Conducted during internship

porates several carefully designed modules—such as Mixture-of-Physical-Experts Attention (MoPA) and a Physical Classifier—to effectively encode these attributes and enhance the model's adherence to physical laws during generation. In addition, most existing video datasets feature only weak or implicit representations of physical phenomena, limiting their utility for learning explicit physical principles. To bridge this gap, we present **WISA-80K**, a new dataset comprising 80,000 human-curated videos that depict 17 fundamental physical laws across three core domains of physics: dynamics, thermodynamics, and optics. Experimental results show that WISA substantially improves the alignment of T2V models (such as CogVideoX and Wan2.1) with real-world physical laws, achieving notable gains on the VideoPhy benchmark. Our data, code, and models are available in the https://wisav1.github.io/WISA/.

# 1 Introduction

Many recent studies (e.g., Cosmos [1], Kling [14], Step-Video-T2V [21], Sora [26], and CogVideoX [41]) have endeavored to develop robust text-to-video (T2V) models for building world simulators [39, 6, 43]. While these models are capable of generating highly realistic and text-consistent videos, leveraging the scale of their data and architectures, they still face challenges in understanding abstract physical principles and producing videos that fully align with real-world physical laws [3, 23].

The substantial gap between abstract physical laws and their visual manifestations presents a significant challenge for injecting physical guidance into T2V models. Physical principles or laws are often conveyed through abstract natural language, reflecting the underlying operational logic of the real world. In contrast, generative models map textual descriptions directly to the visual appearance of objects, including their color and shape. There is a complex logical reasoning process between physical principles and the visual physical phenomena they give rise to. However, generative models, which are trained to map learned data distributions, struggle to extract appropriate physical information from a single textual instruction and translate it into a physically consistent visual representation for a specific scenario. This challenge becomes even more pronounced in video generation, where the strict temporal order of physical events must be preserved.

To this end, we propose the **W**orld **S**imulator **A**ssistant (**WISA**), which decomposes abstract physical principles into multiple categories of physical information and integrates them into T2V models to enable physics-aware generation. Specifically, it decomposes physical principles into three levels: textual physics descriptions, qualitative physics categories, and quantitative physical properties, and designs appropriate tailored injection methods for each type of information. The **Textual Physical Description** outlines the physical principles relevant to the scene, the resulting physical phenomena, and their specific visual manifestations. WISA incorporates this information by concatenating it with the caption before feeding it into the text encoder. The **Qualitative Physics Categories** indicate the types of physical phenomena that may be present in a scene. Following the focus of existing physical T2V benchmarks (e.g., VideoPhy and PhyGenBench), WISA targets 17 representative phenomena commonly encountered in video generation tasks. These span three major branches of physics (i.e., dynamics, thermodynamics, and optics) and include examples such as collision (dynamics), refraction (optics), and melting (thermodynamics). Recognizing that different physical phenomena require distinct physical features, WISA proposes **M**ixture-of-**P**hysical-Experts **A**ttention (**MoPA**), inspired by MoE [30] and MoH [12]. MoPA assigns expert attention heads to individual physics categories, activating only the relevant experts during generation to specialize in modeling the associated phenomena. When a scene involves multiple physical phenomena, MoPA dynamically activates multiple expert heads, allowing the model to effectively capture and synthesize complex physical interactions. **Quantitative Physics Properties** refer to numerical physical attributes that directly influence the physical process, such as density, duration, and temperature. WISA encodes these properties as physical embeddings and injects them into the model via AdaLN [27]. In addition, WISA employs a Physical Classifier, which is designed to recognize qualitative physics categories and assist in perceiving physical properties.

However, extracting the above physical information and subsequently understanding physical principles from general scene video in existing datasets [25, 36] is a suboptimal approach for T2V models. Firstly, general scene videos often feature the interweaving of multiple physical phenomena. Individual physical phenomena are not prominently visualized, which makes it difficult to accurately extract physical information and establish a precise connection between the physical data and its

corresponding visual manifestation. Secondly, in these datasets, only a few videos distinctly highlight specific physical phenomena as representative examples, while most videos treat physical phenomena as secondary elements. For instance, in the Figure 2, the flow of water is a secondary element. Despite having physical information guidance, the T2V models are unable to perceive the physical principles of fluid motion from this type of data.

To address these challenges, we collect and construct **WISA-80K**, a dataset containing **80,000** videos that represent 17 physical phenomena across three major branches of physics as shown in Figure 1, designed as a data assistant for world simulators. Specifically, based on the previously defined physics categories, we manually collect videos that clearly exhibit obvious physical phenomena corresponding to each category (e.g., as shown in the lower part of Figure 2). We then apply shot boundary detection, aesthetic

Liquid motion in *general scenarios*: Physical phenomena are not obvious

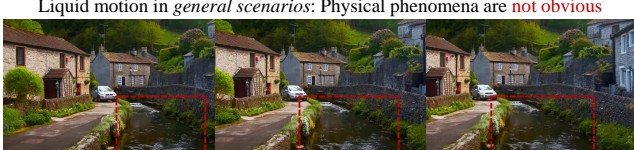

Liquid motion in *WISA-80K*: Physical phenomena are obvious

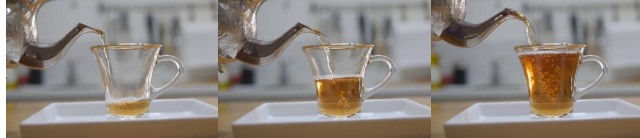

Figure 2: Comparison between general videos in Koala-36M and videos with distinct physical phenomena in WISA-80K.

quality filtering, and video captioning to the raw videos. Subsequently, we leverage GPT-4o mini to extract and decompose the physical information from the video captions into textual physics descriptions, qualitative physics categories, and quantitative physics properties for WISA.

Our contributions can be summarized as follows:

- We propose a physical principle decoupling method, bridging the gap between physical laws and generative modeling. In this method, physical principles are represented as structured physical information, encompassing textual physical descriptions, qualitative physics categories, and quantitative physical properties.

- We present the World Simulator Assistant (WISA), which guides T2V models to efficiently learn specific physical phenomena based on structured physical information, through specialized designs such as Mixture-of-Physical-Experts Attention (MoPA) and Physical Classifier.

- We manually collect 80,000 video clips that clearly showcase physical phenomena, creating the first large-scale physics video dataset, WISA-80K. It broadly covers common physical phenomena observed in the real world, encompassing 17 types of physical events (e.g., Collision, Melting, and Reflection) across three major branches of physics.

- Quantitative and qualitative experimental results demonstrate WISA and WISA-80K can greatly assist basic T2V models in producing videos that better align with real-world physical laws, while introducing only a 3.5% increase in parameter count and 5% inference time.

## 2 Related Work

**Text-to-Video Generation** Early text-to-video (T2V) generation research [10, 9, 33, 7, 8, 34, 4] primarily extend image generation models [5, 18, 17, 28, 19, 22] with temporal capabilities to enable video generation. These methods often suffered from limited realism and restricted motion dynamics. The powerful 3D spatio-temporal modeling and scalability of Diffusion Transformers [27, 15] have greatly advanced the development of visual generation models. Enabled by Diffusion Transformers, a series of recent T2V works (including OpenSora [42], Cosmos [1], Sora [26], CogVideoX [41], HunyuanVideo[x], Kling [14], Wan2.1 [31], and Step-Video-T2V [21]) significantly improve the realism and motion quality of video generation by scaling up model parameters and training data. These works are widely considered as a promising pathway towards building a World Simulator. However, they still struggle to generate videos that fully comply with real-world physical laws as they essentially fit the data distribution [13] from general-scene datasets such as Koala-36M [36] and OpenVid [25], where physical laws are not explicitly reflected and physical phenomena are not prominently presented (e.g., in the upper part of Figure 2). In contrast, our carefully curated WISA-80K dataset prioritizes the explicit presentation of typical physical phenomena as the primary criterion for video collection as presented in Figure 1. And it provides detailed and structured

physical information annotations, making it a valuable data assistant for enhancing the physical consistency of video generation.

**Physical-aware Video Generation** Recently, researchers [24, 3, 23, 16, 2, 20, 40, 37] have increasingly focused on improving and evaluating the physical consistency of generated videos. On the one hand, Videophy [3] and PhyGenBench [23] build test samples that reflect various physical laws, and they evaluate how well generated videos follow real-world physical laws by either training physics classification models with manual annotations or using question-answering methods based on Vision-Language models [38]. Physics-IQ[24] establishes a high-quality image-to-video benchmark designed to evaluate the ability of I2V models to generate physically consistent video sequences based on an initial state and textual instructions. On the other hand, DANO [16], MotionCraft [2], and PhysGen [20] parse objects from images and estimate their rigid motion in a differentiable manner by considering physical properties such as mass, inertia, friction, and rotation. Based on these estimations, they animate the images into videos. However, these methods are restricted to fixed physical categories (e.g., rigid motion) and static scenarios that involve only object motion, which hinders their generalizability. PhyT2V [40] leverages large language models and vision-language models to extract physical inconsistency information from generated videos. Based on the extracted physical feedback, it iteratively refines the textual description over multiple rounds, improving video generation quality. Although this approach offers generality, it introduces significant inference overhead and fails to enhance the generative model's ability to encode physical knowledge. In this paper, WISA incorporates structured physical information into the generative model, enhancing its physical perception and enabling it to handle various physical phenomena more effectively.

## 3 WISA-80K

### 3.1 Data Collection and Annotation

**Physical Laws Definition:** In previous physics evaluation benchmarks [3, 23], the physical phenomena emphasized in video generation tasks have primarily focused on three fundamental branches of physics: *Dynamics*, *Thermodynamics*, and *Optics*. Therefore, in this paper, we select 17 representative physical phenomena from these three core domains, excluding specialized cases such as electromagnetic phenomena.

*Dynamics*: We consider six common dynamic phenomena: *Collision*, *Rigid Body Motion*, *Elastic Motion*, *Liquid Motion*, *Gas Motion*, and *Deformation*. For instance, the swinging of a pendulum serves as an example of *Rigid Body Motion*.

*Thermodynamics*: We select six common thermodynamic phenomena: *Melting*, *Solidification*, *Vaporization*, *Liquefaction*, *Explosion*, and *Combustion*. For example, a time-lapse of melting ice cream illustrates the *Melting* phenomenon.

*Optics*: We define five common optical phenomena: *Reflection*, *Refraction*, *Scattering*, *Interference and Diffraction*, and *Unnatural Light Sources*.

We did not include certain physical phenomena (e.g., sublimation, condensation) due to their infrequent occurrence in real-world scenarios and the associated difficulties in collecting sufficient high-quality data. Subsequently, for each selected physical phenomenon, we manually collected videos from the Internet that clearly demonstrate the corresponding behavior, without relying on any existing video datasets for filtering or selection. During the collection process, we exclude videos with overlaid text or significant visual blur to ensure clarity and quality. As a result, we curate a dataset comprising approximately 40,000 videos.

**Pre-processing and Caption:** We use PySceneDetect [29] to split the raw videos into individual scene clips, followed by filtering based on aesthetic scores. This process yields approximately 80,000 high-quality clips. Then, we utilize Qwen2.5-VL [35] to generate video captions using the following prompt: {*Please describe the content of this video in as much detail as possible, including the objects, scenery, animals, and camera movements within the video.*} The caption length is limited to 256 tokens.

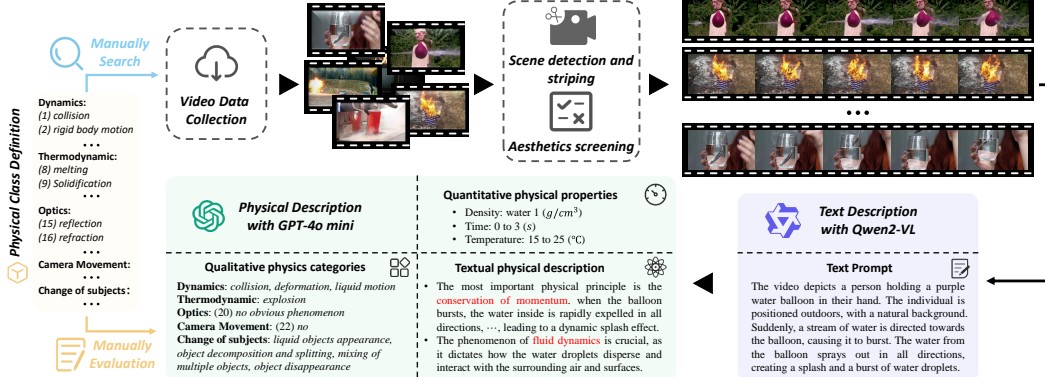

Figure 3: Pipeline of WISA-80K. We first define 17 common physical phenomena and, based on this, manually collect 80,000 video samples that clearly illustrate these phenomena. Then, we perform shot detection and aesthetic filtering on the raw videos. Text descriptions are extracted using Qwen2.5-VL, and detailed physical annotations are generated with GPT-4o mini.

## 3.2 Physical Information Decompose

We believe that simple video captions are not sufficient to clearly represent the physical information and related physical phenomena in a video. As shown in the Figure 3, we further constructed structured physical annotations to analyze the physical information from multiple dimensions. Specifically, we decompose the physical information into: *textual physical descriptions*, *qualitative physics categories*, and *quantitative physical properties*.

**Textual physical descriptions**: Provide a detailed explanation of the physical principles to be considered and the resulting intuitive physical phenomena, while supplementing the missing physical information in the prompt. For instance, the prompt "*an antique clock swings*" corresponds to the textual physical description: "*... the amplitude of the swing gradually decreases ...*".

**Qualitative physics categories**: These indicate the types of physical phenomena involved in a video. Although each video is collected based on a specific physical phenomenon, it may still encompass multiple types. Therefore, for each video, we identify the presence of dynamics-related, optics-related, and thermodynamics-related phenomena to effectively handle cases where multiple physical effects are coupled. Additionally, three categories of anomalies (i.e., *No obvious dynamic phenomenon*, *No obvious thermodynamic phenomenon*, and *No obvious optical phenomenon*) are introduced to account for scenarios that do not involve dynamics, thermodynamics, or optical phenomena. Furthermore, nine categories of visual phenomena are introduced, two of which pertain to whether the shot exhibits motion, while the remaining seven correspond to changes in the state of moving entities (i.e., *Object decomposition and splitting*, *Mixing of multiple objects* ... For detailed explanations, please refer to **Supplementary Material A.5**. In total, there are 29 qualitative physics categories.

**Quantitative physical properties**: Three physical attributes related to multiple physical phenomena are annotated, namely the density of primary motion physics, the time range during which the physical phenomenon occurs, and the temperature range during which the physical phenomenon occurs.

Due to the significant computational overhead and cost associated with video multi-modal models, the annotation of the above physical information is carried out using GPT-4o mini based on the caption. Specifically, we conduct five rounds of annotation to label qualitative physical phenomenon categories (i.e., dynamics, thermodynamics, optics, motion, the state of objects), and three rounds to annotate quantitative physical attributes (i.e., *Density*, *Time* and *Temperature*). Detailed annotation prompts and examples are provided in the **Supplementary Material A.7 and A.8**.

To verify the reliability of the automatic annotations, we conducted a manual evaluation on a randomly sampled subset of 500 videos from our dataset. Our evaluation focused on the three types of AI-generated annotations: 1) Textual Physical Descriptions: We measured human rater satisfaction (i.e., does the description accurately reflect the video's physics?). Result: 95% satisfaction rate. 2) Qualitative Physics Categories: We compared the AI-generated labels against the ground-truth labels assigned during initial video collection. Result: 76% accuracy. 3) Quantitative Physical Properties: We again measured human rater satisfaction (i.e., are the estimated density/time/temperature values plausible?). Result: 86% satisfaction rate. While some label noise is inherent in any large-scale,

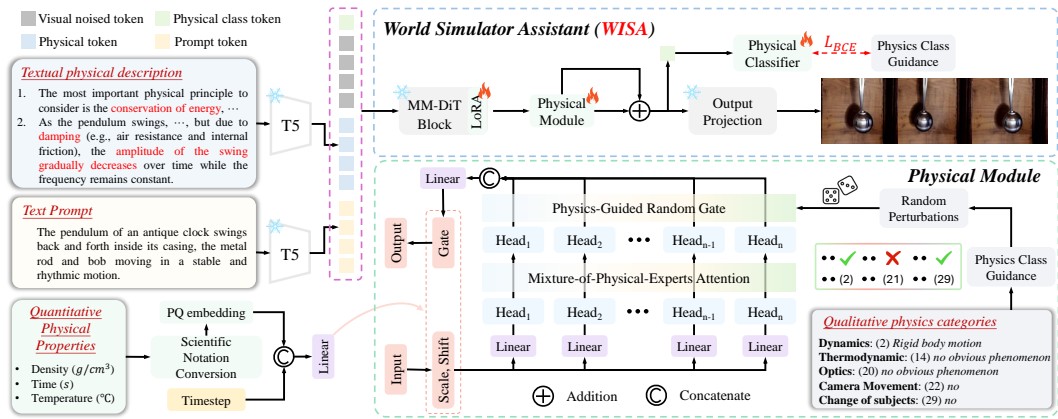

Figure 4: Overview of the proposed WISA. WISA introduces the Physical Module and Physical Classifier, which leverage structured physical annotations to guide and assist T2V models in generating physics-aware videos.

automatically annotated dataset, these results demonstrate that the overall quality of WISA-DATA-80K's annotations is high. The data is sufficiently reliable to provide a strong learning signal, as evidenced by the performance gains in experiments. More analysis of WISA-80K please refer to the **Supplementary Material A.6 and A.9**.

# 4 Method

## 4.1 Overview

Given textual physical descriptions, qualitative physical categories, and quantitative physical properties, we design the WISA framework to efficiently incorporate these conditions into existing T2V models (i.e., CogVideoX [41] or Wan2.1 [31]). To facilitate the learning of physical knowledge while preserving the model's original capabilities with limited video data, we design three distinct condition injection methods tailored to each of the three categories of physical information, as illustrated in Figure 4. Specifically, for the textual physical descriptions, we concatenate them with the video caption and leverage the generative model's inherent semantic understanding to generate visual phenomena described in text (Such as "amplitude of the swing gradually decreases over time" in Figure 4). For qualitative and quantitative physical conditions, WISA introduces the Physical Module. In this module, we propose a Mixture-of-Physical-Experts Attention (MoPA), which assigns expert heads to each physics category to model category-specific features. Quantitative physical quantities are encoded as physical embeddings and then integrated into the denoising feature within the module using AdaLN. Additionally, we introduce a qualitative Physical Classifier to help the model understand the physical conditions. Due to the significant computational and parameter cost introduced by MoPA, only one physical module is inserted after all the Diffusion transformer blocks to accelerate training and reduce the overall burden. Detailed explanations and elaborations of the Physical Module and Physical Classifier are provided in Sec. 4.2 and Sec. 4.3.

## 4.2 Physical Module

Most videos from real-world scenes involve the coupling of multiple physical phenomena. Even when decomposed into distinct physical categories in WISA-80K, it remains challenging for T2V models to comprehend the abstract qualitative physical categories and accurately model specific types of physical phenomena. To address this challenge, we propose a Mixture-of-Physical-Experts Attention within the Physical Module. Inspired by MoH [12], this mechanism assigns each head in the multi-head self-attention to a specific class of physical phenomena and activates the output of the relevant head only when the corresponding phenomenon is present. This approach treats each head as an expert in its domain, enabling it to independently model the properties of a particular physical phenomenon. In the presence of coupled physical phenomena, multiple corresponding expert heads are activated to effectively model the interactions among them.

Specifically, qualitative physical categories are encoded as $P_c \in \mathbb{R}^C$, where $C$ denotes the number of defined physical phenomena (i.e., 29). Here, $P_c^i = 0$ indicates that the corresponding category is not activated, and $P_c^i = 1$ indicates that the corresponding category is activated, with $i$ being the category index. Physical categories cannot be absolutely correct and may contain noise, such as incorrect activations or suppressions. To mitigate the impact of these noises on training, we employ a random perturbation operation, where the positions with $P_c^i = 1$ are set to 0, and the positions with $P_c^i = 0$ are set to 1.0 with a certain probability (i.e., 0.2), resulting $\hat{P}_c$. After the multi-head self-attention operation, the denoising feature $F_h \in \mathbb{R}^{N \times d \times h}$ (where $h$ presents the number of head and $h = C$, and $d$ denotes head dimension) will interact with $\hat{P}_c$ to activate and suppress the experts corresponding to different physical phenomena. The feature dimension is then restored through concatenation and a linear layer. The mathematical representation of this process is as follows:

$$\hat{P}_c = \text{Random}(P_c), \ F_h = \text{MHSA}(F),$$
$$F_o = \text{Linear}(\text{Reshape}(F_h \odot \hat{P}_c)) \tag{1}$$

where $\text{Random}$ denotes random perturbations operation, $\text{MHSA}$ represents multi-head self-attention, and $\odot$ denotes element-wise multiplication.

Due to the large variations in the time and temperature spans of different physical phenomena, we first represent the temperature and time in quantitative information using scientific notation, with coefficients and exponents. These values $P_p \in \mathbb{R}^n$ are mapped through a linear layer, concatenated with the timestep embedding $T_e \in \mathbb{R}^t$, and injected by AdaLN. The mathematical representation of this process is as follows:

$$\alpha, \beta, \gamma = \text{Chunk}(\text{Linear}(\text{Concat}(\text{Linear}(P_p), T_e)), \dim = -1)$$
$$F = F * (1 + \alpha) + \beta, \ F_o = F_o * \gamma \tag{2}$$

Generative models often consist of multiple transformer blocks with large feature dimensions, inserting the Physical Module after every block would lead to an explosion in both parameters and computational complexity. Therefore, we insert the Physical Module only after the final transformer block, achieving efficient physical information guidance while mitigating the aforementioned issues.

### 4.3 Physical Classifier

To guide the generative model in understanding abstract physical categories and modeling physical properties, we introduce a Physical Classifier after the Physical Module to predict qualitative physical categories. We introduce a learnable embedding vector, which we call the [PHYSICS_TOKEN]. This token is concatenated with the noisy visual tokens and text prompt tokens and is processed by the entire MM-DiT and MoPA architecture. The final hidden state of the [PHYSICS_TOKEN] $F_c \in \mathbb{R}^C$. is fed into a simple MLP classification head. This head outputs logits corresponding to the 29 qualitative physical categories, performing a multi-label classification task over the categories defined in our work. This output is used only to compute the multi-label binary cross-entropy loss for training.

$$L_{pc} = \sum_{i=1}^{C}(P_c^i \log(f_c^i) + (1 - P_c^i)\log(1 - f_c^i)), \tag{3}$$

where $C$ is the number of physical categories, and $f_c \in \mathbb{R}^C$ represents the predicted probabilities, which are obtained by passing $F_c$ through the sigmoid function. For each video, the model predicts which of these phenomena are present. During inference, the output from the classifier head is entirely discarded and has no influence on the video generation process. Its sole purpose is to serve as an auxiliary training signal, guiding the model to better learn and represent physical concepts.

To balance the introduced classification loss $L_{pc}$ and the diffusion loss $L_{diffusion}$, we adopt the following loss function to optimize the physics-aware generative model.

$$L = L_{diffusion} + \lambda L_{pc}/(1 + L_{pc}.\text{detach}), \tag{4}$$

where $\lambda$ is balance coefficient.

Table 1: Quantitative evaluation using VideoCon-Physics conduct on the Videophy and PhyGenBench prompt lists. The best performing metrics are highlighted in **bold**.

| Method | Inference Time (s) | Prompts from VideoPhy [3] | | | | Prompts from PhyGenBench [23] | | | |
|---|---|---|---|---|---|---|---|---|---|
| | | IS (↑) | CLIPSIM (↑) | SA (↑) | PC (↑) | IS (↑) | CLIPSIM (↑) | SA (↑) | PC (↑) |
| VideoCrafter2 [7] | - | - | - | 0.47 | 0.36 | - | - | - | - |
| OpenSora [42] | - | 28.72 | 0.2638 | 0.21 | 0.35 | - | - | - | - |
| HunyuanVideo [17] | - | - | - | 0.46 | 0.28 | - | - | - | - |
| Cosmos-Diffusion-7B [1] | 600 | 25.58 | 0.2444 | 0.52 | 0.27 | 20.17 | 0.1956 | 0.41 | 0.24 |
| CogVideoX-5B [41] | 210 | 30.17 | 0.2714 | 0.57 | 0.41 | 26.49 | 0.2590 | 0.34 | 0.42 |
| CogVideoX-5B + PhyT2V [40] | 1800 | - | - | 0.59 | 0.42 | - | - | 0.38 | 0.42 |
| CogVideoX-5B-WISA | 220 | 34.62 | **0.2822** | **0.62** | **0.45** | 27.31 | **0.2813** | 0.39 | **0.45** |
| Wan2.1-14B [31] | 900 | 36.52 | 0.2686 | 0.54 | 0.31 | 33.41 | 0.2488 | 0.39 | 0.28 |
| Wan2.1-14B-WISA | 960 | **38.18** | 0.2813 | 0.60 | 0.36 | **37.62** | 0.2725 | **0.42** | 0.33 |

## 5    Experiments

**Training Setting:**    We select the current representative open-source T2V model, CogVideoX-5B and Wan2.1-14B, as the base T2V models to validate the effectiveness of WISA. More training detail, please refer to **Supplementary Material A.3**.

**Evaluation:**    We select VideoCon-Physics from Videophy [3] to evaluate the physical law consistency (PC) and semantic coherence (SA) of the generated videos. We use 160 carefully crafted prompts from PhyGenBench [23] and 344 prompts from Videophy, designed to reflect various physical principles, for testing. Following VideoCon-Physics[2], we compute SA and PC by averaging the predicted results. Additionally, we adopt the Inception Score (IS) to evaluate the perceptual quality of generated videos and employ CLIP similarity (CLIPSIM) [11] to measure text-video alignment. More evaluation detail, please refer to **Supplementary Material A.3**.

### 5.1    Quantitative comparison

We select five general text-to-video generation models (i.e., VideoCrafter2, OpenSora, HunyuanVideo, CogVideoX-5B and Cosmos-Diffusion-7B) and PhyT2V, a method specifically designed to enhance physical properties, for quantitative comparison, as shown in Table 1.

**VideoPhy**: WISA achieves state-of-the-art performance on both SA and PC metrics while maintaining high efficiency. Compared to the CogVideoX-5B, CogVideoX-5B-WISA improves SA and PC scores by 0.05 and 0.04, respectively, demonstrating that our proposed method significantly enhances the realism of generated videos. PhyT2V improves its performance by iteratively analyzing physical errors in generated video captions and adjusting the input prompts based on feedback from VideoCon-Physics scores. However, its cumbersome pipeline, which involves multiple rounds of Tarsier-34B [32] inference for video generation, introduces extremely long inference time—approximately 9 times longer than the original generation model. Cosmos exhibits poor performance due to the disordered physical processes and inconsistent temporal sequences. Furthermore, Wan2.1-14B-WISA also achieves performance improvements on Wan2.1, showing advantages in metrics such as IS and CLIPSIM. However, since Wan2.1 is not among the nine generative models used to construct the training data for VideoCon-Physics (whereas CogVideoX is included), it shows certain disadvantages in SA and PC. Despite this, Wan2.1 demonstrates superior video quality, achieving better performance in the IS.

**PhyGenBench**: We also evaluate WISA using prompts from PhyGenBench, observing significant improvements on both CogVidoX and Wan2.1, which demonstrates the generalizability of WISA.

### 5.2    Qualitative comparison

We further provide a qualitative comparison with existing methods to demonstrate the advantages of WISA. As shown in the Figure 5, for the example of the rope supports a wooden swing, CogVideoX-

---

[2]https://github.com/Hritikbansal/videophy/issues/5

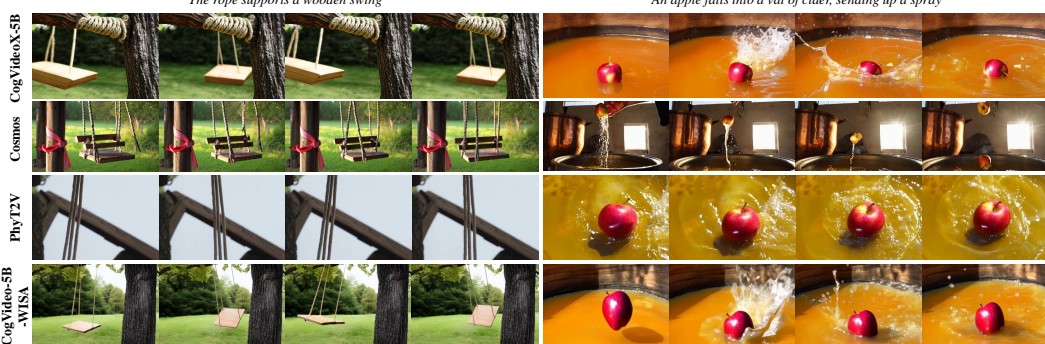

Figure 5: Qualitative comparison between CogVideoX-5B-WISA and existing T2V methods. CogVideoX-5B-WISA exhibits better alignment with real-world physical laws.

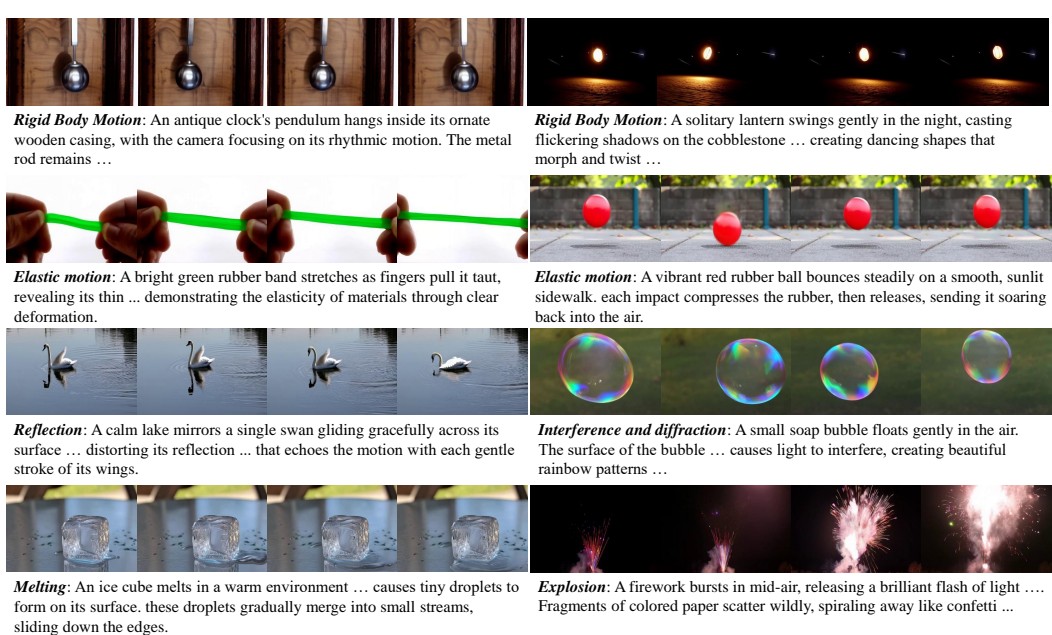

*Rigid Body Motion*: An antique clock's pendulum hangs inside its ornate wooden casing, with the camera focusing on its rhythmic motion. The metal rod remains …

*Rigid Body Motion*: A solitary lantern swings gently in the night, casting flickering shadows on the cobblestone … creating dancing shapes that morph and twist …

*Elastic motion*: A bright green rubber band stretches as fingers pull it taut, revealing its thin ... demonstrating the elasticity of materials through clear deformation.

*Elastic motion*: A vibrant red rubber ball bounces steadily on a smooth, sunlit sidewalk. each impact compresses the rubber, then releases, sending it soaring back into the air.

*Reflection*: A calm lake mirrors a single swan gliding gracefully across its surface … distorting its reflection ... that echoes the motion with each gentle stroke of its wings.

*Interference and diffraction*: A small soap bubble floats gently in the air. The surface of the bubble … causes light to interfere, creating beautiful rainbow patterns …

*Melting*: An ice cube melts in a warm environment … causes tiny droplets to form on its surface. these droplets gradually merge into small streams, sliding down the edges.

*Explosion*: A firework bursts in mid-air, releasing a brilliant flash of light …. Fragments of colored paper scatter wildly, spiraling away like confetti ...

Figure 6: More samples generated by CogVideoX-5B-WISA, covering additional physical phenomena.

5B-WISA generates a video where the rope suspending the wooden seat swinging back and forth in accordance with physical laws. In contrast, CogVideoX-5B produces unstable swing motion; PhyT2V fails to generate the swinging behavior of the wooden seat; and Cosmos generates a physically inconsistent scene where the rope breaks while the wooden seat remains horizontally suspended. In the example on the right, WISA successfully simulates the process of an apple falling into water: the water surface remains calm before the apple enters, splashes form as the apple impacts the water, and the apple experiences buoyant force after submersion. However, CogVideoX-5B generates chaotic water and apple movements, PhyT2V omits the falling process, and Cosmos mistakenly generates two apples at the end. Additional videos generated by CogVideoX-5B-WISA, demonstrating various physical phenomena, are also presented in the Figure 6. All aforementioned videos, along with comparisons on Wan2.1, are provided in the Project Page.

## 5.3 Ablation Study

We conduct ablation studies on VideoPhy using VideoCon-Physics to verify the effectiveness of key components in our method, as shown in the Table 2. The baseline is CogVideoX-5B. As expected, removing MoPA results in a performance drop due to the absence of qualitative physical information as guidance. Similarly, the inclusion of the Physical Classifier aids the generative model in perceiving and modeling physical properties, thereby enhancing both semantic relevance and consistency with physical laws. Notably, the evaluation model VideoCon-Physics [3] is trained on samples generated

by nine different T2V models, leading to a distribution shift when compared to the real-world videos in WISA-80K. Consequently, relying solely on LoRA yields only limited improvement.

To further investigate the impact of clearly-defined physical phenomena data versus general scene data on physical perception, we fine-tune LoRA on 80,000 videos from an open-source video dataset. This results in only a slight performance decline, indicating that the physically grounded videos in WISA-80K provide substantial value for modeling physical properties.

Table 2: Ablation study on the key components of WISA. "PC" denotes the Physical Classifier.

| Setting | Data | Textual Physical Descriptions | Qualitative Physics Categories | Quantitative physical properties | SA (↑) | PC (↑) |
|---------|------|------------------------------|-------------------------------|----------------------------------|--------|--------|
| Baseline | - | - | - | - | 0.57 | 0.41 |
| only LoRA | General Data | - | - | - | 0.57 | 0.40 |
| only LoRA | WISA-80K | ✓ | - | - | 0.58 | 0.43 |
| w/o MoPA | WISA-80K | ✓ | - | ✓ | 0.59 | 0.43 |
| w/o MoPA | WISA-80K | - | - | ✓ | 0.57 | 0.42 |
| w/o PC | WISA-80K | - | ✓ | - | 0.60 | 0.44 |
| w/o PC | WISA-80K | ✓ | ✓ | ✓ | 0.61 | 0.44 |
| WISA | WISA-80K | ✓ | ✓ | ✓ | 0.62 | 0.45 |

## 5.4 User Preference

The physical consistency of generated videos is abstract and difficult to quantify directly. Therefore, we conduct a human evaluation to assess the effectiveness of WISA. Specifically, we selected three representative models for comparison. The evaluation considered two aspects: semantic consistency and physical alignment. Each candidate model is ranked in both aspects, receiving a score based on its ranking: 3 points for first place, 2 points for second, and 0 points for last. We collected preference results from 100 participants. As shown in the Figure 7 demonstrates that WISA achieves a significant advantage in physical alignment, while also maintaining strong semantic consistency.

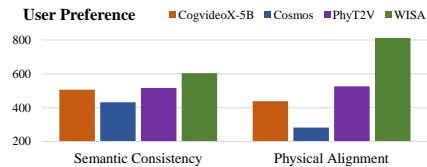

Figure 7: User Preference on VideoPhy prompts.

## 5.5 Attention Map Analysis

We further conduct a visual analysis of the Mixture-of-Physical-Experts attention maps, aiming to investigate whether different physical experts focus on the regions corresponding to distinct physical phenomena. As shown in the Figure 8, the rigid body motion expert perfectly focuses on the swing region, while the non-dynamics expert attends to the static background with no apparent motion. This demonstrates that the MoPA effectively models and captures the corresponding physical attributes.

## 6 Conclusion

In this paper, we present WISA framework, which decomposes physical principles into structured physical information, including textual physical descriptions, qualitative physical categories, and quantitative physical properties. To help T2V models learn these physical aspects effectively, WISA incorporates two key components: the Mixture-of-Physical-Experts Attention and the Physical Classifier. Building on this, we construct WISA-80K, a dataset containing 80,000 video clips that cover 17 physical phenomena across three fundamental categories of physics, providing a high-quality data foundation. Experimental results show that WISA and WISA-80K can effectively help produce videos that

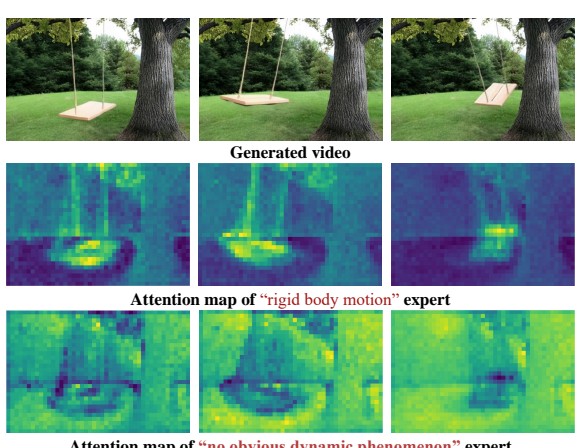

Figure 8: Attention maps of different physical experts.

better align with real-world physical laws, while the additional computational overhead is under 5%. We hope that WISA can provide valuable insights into the research on building powerful world simulators. We further discuss the limitation of this paper in the Supplementary Material A.2.

## ACKNOWLEDGEMENTS

This work is supported by Scientific Research Innovation Capability Support Project for Young Faculty (No.ZYGXQNJSKYCXNLZCXM-I28), National Natural Science Foundation of China (NSFC) under Grants No.62476293 and General Embodied AI Center of Sun Yat-sen University.

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

# A    Technical Appendices and Supplementary Material

## A.1    Broader Impacts and Ethical Considerations

The development of powerful generative models, particularly those capable of creating realistic video content, carries a significant responsibility. The work presented in this paper, while aimed at advancing scientific understanding and technical capability in physics-aware video generation, is not exempt from potential societal impacts and ethical challenges. This section transparently discusses these issues and outlines the steps we are taking to mitigate potential harms.

**Potential for Misuse and Societal Impact:**    We recognize the dual-use nature of our work. The same technology that can be used for creative applications, scientific simulation, or special effects could also be exploited to create compelling but fabricated content for malicious purposes. The enhanced physical realism of videos generated by a model like WISA could amplify the verisimilitude of synthetic media, increasing its potential to be used for misinformation, disinformation (e.g., "deepfakes"), or propaganda. Such content could erode public trust, be used as false evidence, or create realistic depictions of accidents or violence to incite fear.

**Dataset Ethics and Release Strategy:**    The creation and distribution of any large-scale dataset require careful consideration of privacy, copyright, and consent. The WISA-80K dataset is constructed using video clips from publicly available channels on YouTube.

- Copyright and Terms of Service: To respect the rights of content creators and to comply with platform Terms of Service, we have adopted a metadata-only release strategy. We will not host or distribute any video files, clips, or raw pixel data. The released dataset will contain only the public YouTube video IDs, the relevant start and end timestamps of the physical phenomena, and our corresponding physical annotations. This is a standard and widely accepted practice in the research community (e.g., AudioSet) that enables reproducible research while avoiding copyright infringement.

- Responsible Data Access: To further ensure responsible use, the WISA-80K metadata will not be available for direct public download. Instead, we will implement a gated access mechanism. Researchers wishing to use the dataset must submit a request outlining their institutional affiliation and research purpose. They will be required to agree to a Data Usage Agreement (DUA), which will stipulate that: (1) the dataset is to be used for non-commercial research purposes only; (2) users are responsible for their own adherence to YouTube's Terms of Service when accessing the original videos; and (3) the metadata and any derived video content may not be redistributed.

- Content Creator Rights: We will provide a clear and accessible opt-out mechanism on our future project page. Any content creator can request the removal of their video's ID from our dataset at any time, and we are committed to promptly honoring all such requests.

**Annotation Process: Quality and Bias:**    The physical annotations in WISA-80K were generated using a large language model (GPT-4o-mini). We acknowledge that this automated process may introduce biases or factual inaccuracies ("hallucinations").

- Human Validation: To quantify the quality and reliability of these annotations, we conducted a human validation study on a randomly sampled subset of 500 videos. The results (detailed in Appendix X) indicate a high degree of quality, with 95% satisfaction for textual descriptions and 86% satisfaction for the plausibility of quantitative estimates. The accuracy for qualitative category classification was 76% against human-assigned labels.

- Dataset Positioning: These results confirm that while some label noise is inherent, the annotations provide a strong and reliable learning signal. Nonetheless, WISA-80K should be understood as a large-scale, weakly-supervised dataset rather than a gold-standard, error-free resource. We encourage future work to further refine and build upon these annotations.

**Mitigation Strategies for the Generative Model:**    To address the risks of misuse associated with the generative model itself, we commit to the following safeguards for any future public release:

- Visible Watermarking: All video outputs generated by our released model will be programmatically embedded with a clear and persistent visible watermark to identify them as synthetic.

- Responsible AI License: We plan to release the model under a Responsible AI License (e.g., a CreativeML Open RAIL-M license). Such licenses contractually prohibit users from employing the model for malicious, deceptive, illegal, or unethical purposes, including the generation of harmful misinformation.

## A.2 Limitation

Although our approach significantly improves the ability of existing T2V models to generate videos that align with real-world physical laws, it still has the following limitations: 1) **Limited physical categories**: We collect 80,000 videos in WISA-80K, covering 17 types of physical phenomena. However, due to constraints in time and manpower, the dataset does not include all physical phenomena encountered in the real world, such as corrosion or vacuum environments. 2) **Limited physical information guidance**: WISA primarily provides high-level semantic guidance and lacks detailed constraints at the physical mechanism level (e.g., energy conservation, Newton's laws). However, introducing more detailed physical principle constraints currently requires modeling object motion based on image or 3D information, which suffers from poor generalization and can only handle limited categories and scenarios. How to incorporate physical principle constraints into text-to-video generation while maintaining generalization remains an area worth further research. 3) **Expanding WISA to Unseen Classes**: We conducted additional qualitative evaluations on unseen physical categories such as corrosion and electromagnetism, as shown in Figure 9. As expected, both our WISA-enhanced model and the base model struggle to generate physically plausible videos for these categories. This limitation primarily stems from the absence of relevant concepts and visual examples in the WISA-80K training set, underscoring the current challenge of generalizing to entirely out-of-distribution physical phenomena. However, we argue that WISA's modular design, particularly the Mixture-of-Physical-Experts Attention (MoPA) mechanism, makes it inherently more scalable than monolithic architectures. When introducing a new physical category, traditional fine-tuning methods (e.g., applying LoRA on the base model) typically require retraining large portions of the network to assimilate the new knowledge. In contrast, MoPA allows for a more targeted and efficient expansion: we can simply add and train a new expert head dedicated to the novel phenomenon while keeping the existing experts largely frozen, thus preserving prior knowledge and facilitating incremental learning.

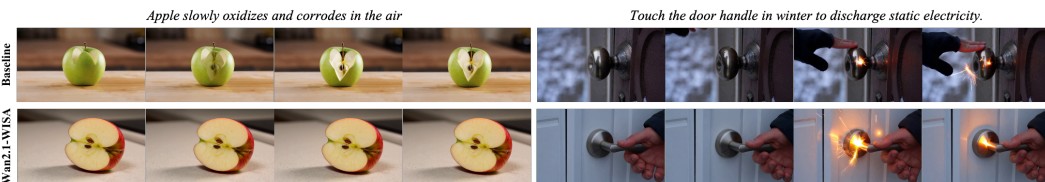

Figure 9: Visual comparison on unseen categories.

**Further work:** In future research, we plan to explore physically consistent generation for the image-to-video (I2V) task. In principle, WISA can be applied to I2V as well. The main distinction between I2V and text-to-video (T2V) lies in the input: I2V models receive a clean latent code for the first frame, whereas the rest of the input text and model architecture remain the same. With explicit initial frame information, the I2V model primarily learns to infer subsequent states, generating intermediate and final frames that follow plausible physical processes.

Compared with T2V, I2V is particularly relevant in embodied intelligence and robotics, where agents often observe a concrete initial state and must predict or plan the outcomes of actions in a physically consistent way. While T2V remains a more challenging task—requiring the model to imagine the entire physical process from scratch based solely on abstract text—the I2V task provides a complementary and practically important setting to study and improve physical consistency in video generation. We aim to extend WISA to I2V in future work, leveraging its modular design to generate videos that maintain accurate physical dynamics starting from a known initial state.

## A.3 Training and Evaluation Detail

**Training:** We choose two representative open-source T2V models—CogVideoX-5B and Wan2.1-14B—as the base models to validate the effectiveness of the proposed WISA. WISA is trained on our constructed WISA-80K dataset for 8,000 steps, using a learning rate of 2e-5 and a batch size of 16. For CogVideoX-5B, the video resolution is set to 480×720 with 49 frames per video, while for Wan2.1-14B, the resolution is 480×832 with 81 frames. We adopt LoRA with a rank of 128 and an alpha of 16. During training, only the physical module, physical classifier, and LoRA parameters are updated, resulting in a total of 187 million learnable parameters for CogVideoX-5B and 587 million for Wan2.1-14B. All experiments are conducted on 8 A100 GPUs, each equipped with 80 GB of memory.

**Evaluation:** VideoCon-Physics was trained by collecting videos generated from nine different models, which were manually annotated for adherence to real-world physical laws and strong semantic consistency. Using this data, a vision-language model (VLM) was fine-tuned to serve as a reward model. During inference, the generated video and corresponding text prompt are fed into VideoCon-Physics, which outputs scores ranging from 0 to 1 for both metrics.

## A.4 Inference without Annotation

For any given user prompt, we use a large language model (GPT-4o) with a set of predefined instructions to generate the required physical annotations (textual description, qualitative categories, quantitative properties). Crucially, this process uses only the input text prompt, with no access to any visual information, thus preventing any unfair information leakage. The cost is also minimal (approx. 2000 tokens per prompt). The detailed instructions are provided in the Figure 12, Figure 13, and Figure 14.

## A.5 The Definition of Physical Categories

We define a total of 29 qualitative physical categories, organized into 5 major classes. The physical categories within each class, along with their corresponding category IDs, are listed as follows:

**Dynamics**: *1. Collision*, *2. Rigid Body Motion*, *3. Elastic Motion*, *4. Liquid Motion*, *5. Gas Motion*, *6. Deformation*, and *7. No obvious dynamic phenomenon*

**Thermodynamics**: *8. Melting*, *9. Solidification*, *10. Vaporization*, *11. Liquefaction*, *12. Explosion*, *13. Combustion* and *14. No obvious thermodynamic phenomenon*

**Optics**: *15. Reflection*, *16. Refraction*, *17. Scattering*, *18. Interference and Diffraction*, *19. Unnatural Light Sources*, and *20. No obvious optical phenomenon*

**Camera motion**: *21. Yes*, *22. No*

**The state of object**: *23. Liquids Objects Appearance*, *24. Solid Objects Appearance*, *25. Gas Objects Appearance*, *26. Object decomposition and splitting*, *27. Mixing of Multiple Objects*, *28. Object Disappearance* and *29. No Change*

Specifically, *Liquids objects appearance*: new liquids appear from the camera over time and due to external forces, such as water squeezed out of a towel. *Solid objects appearance*: new solids appear from the camera over time and due to external forces, such as Chemical reaction that produces precipitates, or cars drive in from outside the camera. *Gas objects appearance*: new gas appears from the camera over time and due to external forces. *Object decomposition and splitting*: Over time and under the action of external forces, an object is broken into multiple sub-parts: such as fruits and vegetables being cut in half. *Mixing of multiple objects*: Over time and with the action of external forces, two objects of the same state mix together, such as two solutions mixing. *Object disappearance*: As time passes and external forces act, objects disappear from the camera. *No change*: No change in the state of the object

## A.6 Dataset Property Analysis

We visualize the distribution of different physics categories and video frame counts in WISA-80K, as shown in the paper Figure 1. Dynamics frequently occur in daily life, accounting for the

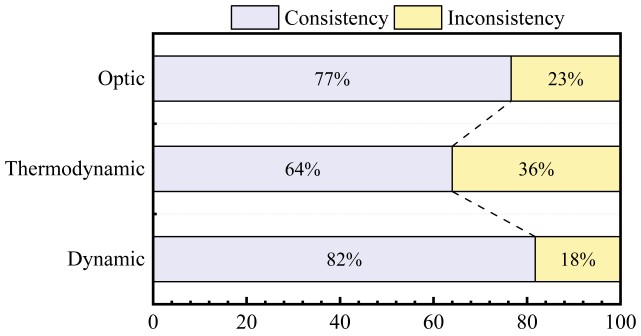

Figure 10: Accuracy of qualitative physical category annotations.

largest proportion at 47%. Optics and thermodynamics, which typically require specific temperature or environmental conditions, account for 29% and 24%, respectively. The proportions of each subcategory are shown in the outer ring of the Figure Based on the labels of the manually collected videos, we evaluate the accuracy of the qualitative physical category annotations. The results are shown in the Figure 10, where the accuracy for dynamics, optics, and thermodynamics reaches 84%, 71%, and 64%, respectively, with an overall accuracy of 75%.

## A.7    More Examples and Annotation

Following the proposed physical information annotation pipeline, we construct the WISA-80K dataset. Several example videos and their corresponding annotations are shown in the Figure 11. This pipeline enables accurate and detailed annotation of physical information, ensuring that each video is comprehensively labeled with its relevant physical properties and phenomena.

## A.8    Annotation Prompts

The detailed prompt used for physical information annotation is illustrated in the Figure 12, Figure 13, and Figure 14.

## A.9    Word Cloud

We conducted a word frequency analysis on the textual physical description in the dataset and generated the word cloud shown in Figure 15. To filter out irrelevant words, we retained only nouns and selected them based on their frequency, from highest to lowest. Notably, physical terms such as 'motion,' 'phenomenon,' and 'light' appear more frequently, highlighting the strong physical relevance of the dataset.

## A.10    Discussion of Quantitative Evaluation

During the quantitative evaluation, we observe several misjudgments in VideoCon-Physics, as shown in the Figure 16. Specifically, WISA generates a physically plausible process where the object enters the water first, followed by the splash, aligning well with real-world physical laws. However, this sample only receives a low score of 0.08 from VideoCon-Physics. We further conduct a simple test using Qwen2.5-VL for evaluation, and the model also struggles to distinguish the correct or incorrect sequence of physical events. These findings show the limitations of existing video-based physics evaluation metrics, indicating that future research into more reliable physical property assessments for videos is necessary.

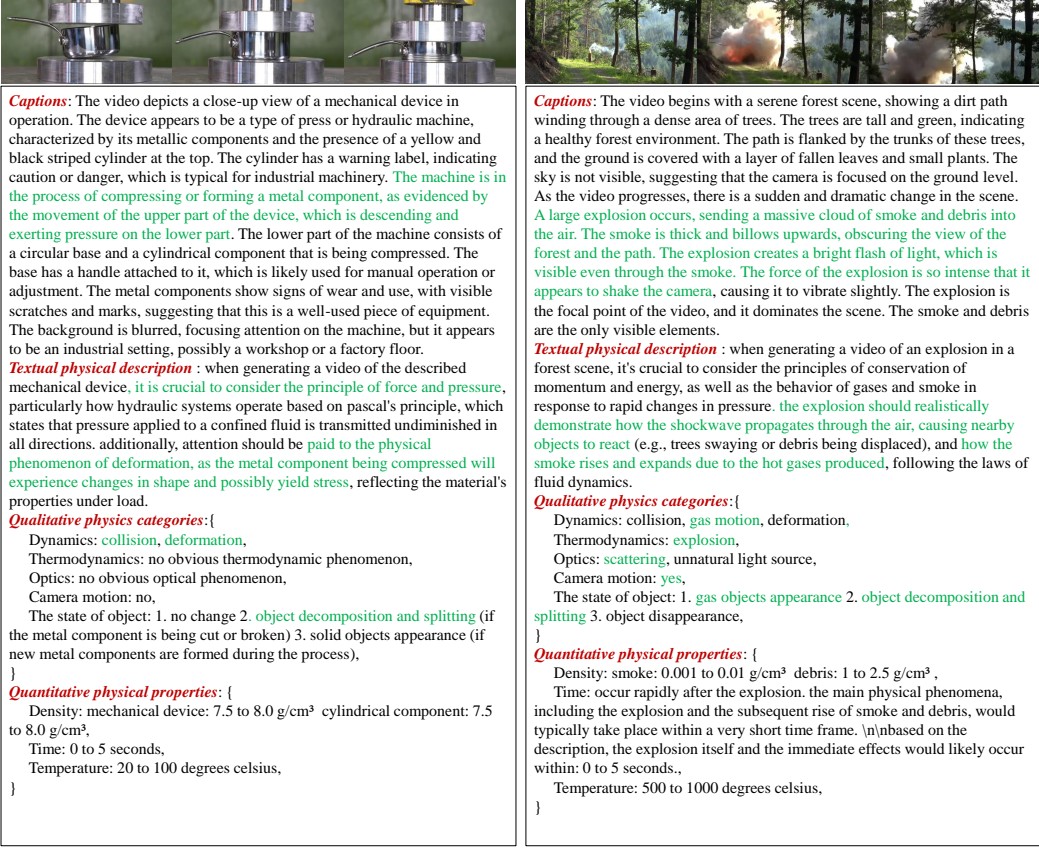

**Captions**: The video depicts a close-up view of a mechanical device in operation. The device appears to be a type of press or hydraulic machine, characterized by its metallic components and the presence of a yellow and black striped cylinder at the top. The cylinder has a warning label, indicating caution or danger, which is typical for industrial machinery. The machine is in the process of compressing or forming a metal component, as evidenced by the movement of the upper part of the device, which is descending and exerting pressure on the lower part. The lower part of the machine consists of a circular base and a cylindrical component that is being compressed. The base has a handle attached to it, which is likely used for manual operation or adjustment. The metal components show signs of wear and use, with visible scratches and marks, suggesting that this is a well-used piece of equipment. The background is blurred, focusing attention on the machine, but it appears to be an industrial setting, possibly a workshop or a factory floor.

**Textual physical description** : when generating a video of the described mechanical device, it is crucial to consider the principle of force and pressure, particularly how hydraulic systems operate based on pascal's principle, which states that pressure applied to a confined fluid is transmitted undiminished in all directions. additionally, attention should be paid to the physical phenomenon of deformation, as the metal component being compressed will experience changes in shape and possibly yield stress, reflecting the material's properties under load.

**Qualitative physics categories**:{
  Dynamics: collision, deformation,
  Thermodynamics: no obvious thermodynamic phenomenon,
  Optics: no obvious optical phenomenon,
  Camera motion: no,
  The state of object: 1. no change 2. object decomposition and splitting (if the metal component is being cut or broken) 3. solid objects appearance (if new metal components are formed during the process),
}

**Quantitative physical properties**: {
  Density: mechanical device: 7.5 to 8.0 g/cm³ cylindrical component: 7.5 to 8.0 g/cm³,
  Time: 0 to 5 seconds,
  Temperature: 20 to 100 degrees celsius,
}

**Captions**: The video begins with a serene forest scene, showing a dirt path winding through a dense area of trees. The trees are tall and green, indicating a healthy forest environment. The path is flanked by the trunks of these trees, and the ground is covered with a layer of fallen leaves and small plants. The sky is not visible, suggesting that the camera is focused on the ground level. As the video progresses, there is a sudden and dramatic change in the scene. A large explosion occurs, sending a massive cloud of smoke and debris into the air. The smoke is thick and billows upwards, obscuring the view of the forest and the path. The explosion creates a bright flash of light, which is visible even through the smoke. The force of the explosion is so intense that it appears to shake the camera, causing it to vibrate slightly. The explosion is the focal point of the video, and it dominates the scene. The smoke and debris are the only visible elements.

**Textual physical description** : when generating a video of an explosion in a forest scene, it's crucial to consider the principles of conservation of momentum and energy, as well as the behavior of gases and smoke in response to rapid changes in pressure. the explosion should realistically demonstrate how the shockwave propagates through the air, causing nearby objects to react (e.g., trees swaying or debris being displaced), and how the smoke rises and expands due to the hot gases produced, following the laws of fluid dynamics.

**Qualitative physics categories**:{
  Dynamics: collision, gas motion, deformation,
  Thermodynamics: explosion,
  Optics: scattering, unnatural light source,
  Camera motion: yes,
  The state of object: 1. gas objects appearance 2. object decomposition and splitting 3. object disappearance,
}

**Quantitative physical properties**: {
  Density: smoke: 0.001 to 0.01 g/cm³ debris: 1 to 2.5 g/cm³ ,
  Time: occur rapidly after the explosion. the main physical phenomena, including the explosion and the subsequent rise of smoke and debris, would typically take place within a very short time frame. \n\nbased on the description, the explosion itself and the immediate effects would likely occur within: 0 to 5 seconds.,
  Temperature: 500 to 1000 degrees celsius,
}

Figure 11: The video data and its detailed annotations in WISA-80K.

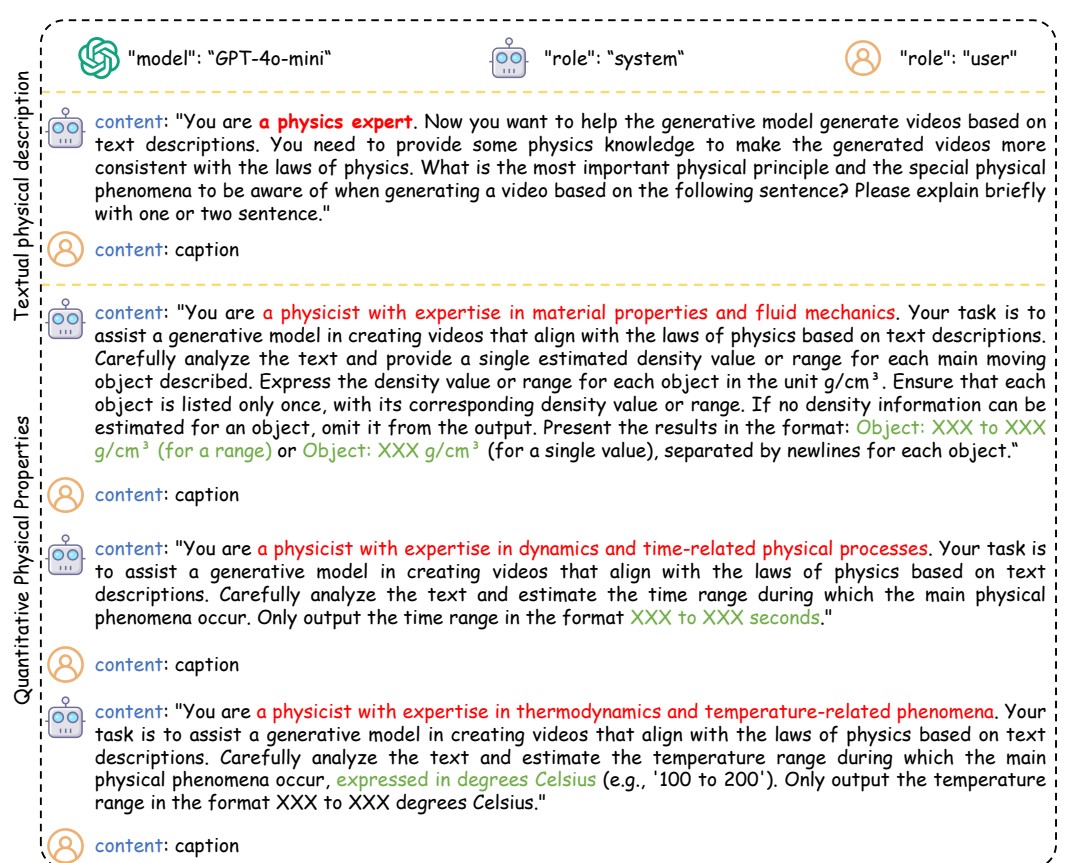

Figure 12: Prompts for annotating textual physical descriptions and quantitative physical properties

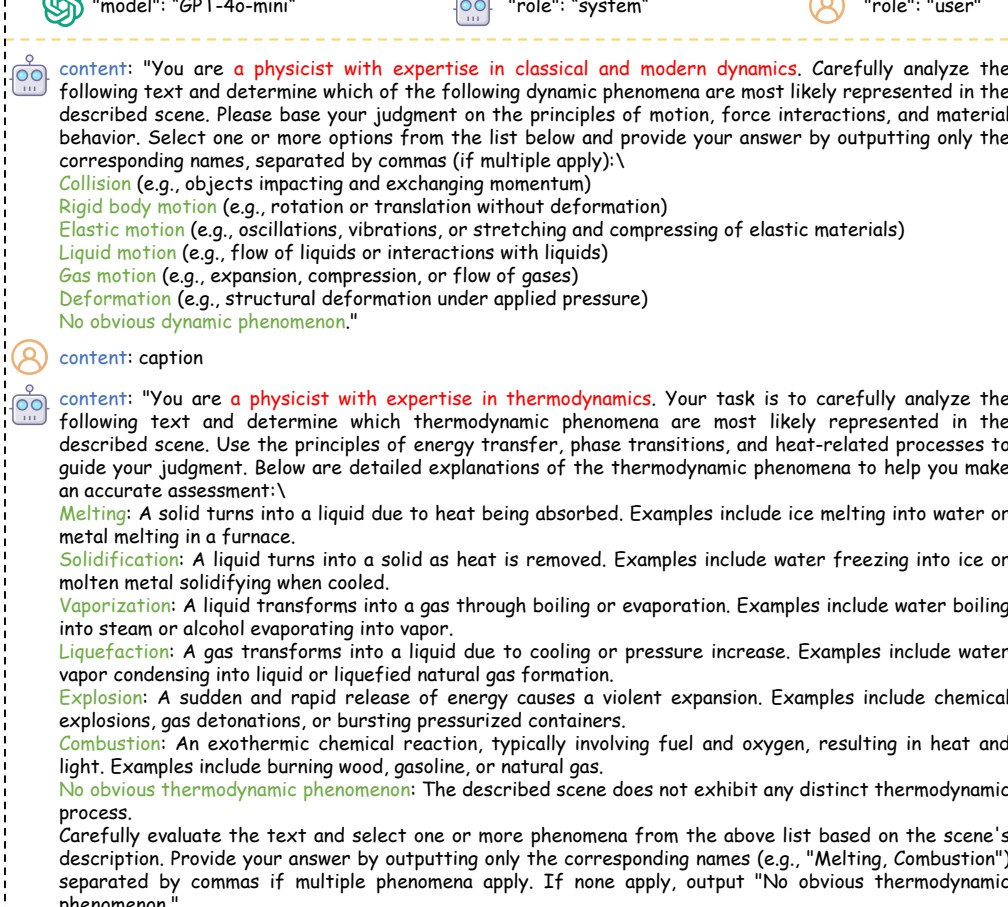

**Qualitative physics categories**

🤖 **"model": "GPT-4o-mini"**   🤖 **"role": "system"**   👤 **"role": "user"**

🤖 content: "You are a physicist with expertise in classical and modern dynamics. Carefully analyze the following text and determine which of the following dynamic phenomena are most likely represented in the described scene. Please base your judgment on the principles of motion, force interactions, and material behavior. Select one or more options from the list below and provide your answer by outputting only the corresponding names, separated by commas (if multiple apply):\
Collision (e.g., objects impacting and exchanging momentum)
Rigid body motion (e.g., rotation or translation without deformation)
Elastic motion (e.g., oscillations, vibrations, or stretching and compressing of elastic materials)
Liquid motion (e.g., flow of liquids or interactions with liquids)
Gas motion (e.g., expansion, compression, or flow of gases)
Deformation (e.g., structural deformation under applied pressure)
No obvious dynamic phenomenon."

👤 content: caption

🤖 content: "You are a physicist with expertise in thermodynamics. Your task is to carefully analyze the following text and determine which thermodynamic phenomena are most likely represented in the described scene. Use the principles of energy transfer, phase transitions, and heat-related processes to guide your judgment. Below are detailed explanations of the thermodynamic phenomena to help you make an accurate assessment:\
Melting: A solid turns into a liquid due to heat being absorbed. Examples include ice melting into water or metal melting in a furnace.
Solidification: A liquid turns into a solid as heat is removed. Examples include water freezing into ice or molten metal solidifying when cooled.
Vaporization: A liquid transforms into a gas through boiling or evaporation. Examples include water boiling into steam or alcohol evaporating into vapor.
Liquefaction: A gas transforms into a liquid due to cooling or pressure increase. Examples include water vapor condensing into liquid or liquefied natural gas formation.
Explosion: A sudden and rapid release of energy causes a violent expansion. Examples include chemical explosions, gas detonations, or bursting pressurized containers.
Combustion: An exothermic chemical reaction, typically involving fuel and oxygen, resulting in heat and light. Examples include burning wood, gasoline, or natural gas.
No obvious thermodynamic phenomenon: The described scene does not exhibit any distinct thermodynamic process.
Carefully evaluate the text and select one or more phenomena from the above list based on the scene's description. Provide your answer by outputting only the corresponding names (e.g., "Melting, Combustion") separated by commas if multiple phenomena apply. If none apply, output "No obvious thermodynamic phenomenon."

👤 content: caption

Figure 13: Prompts for annotating qualitative physics categories

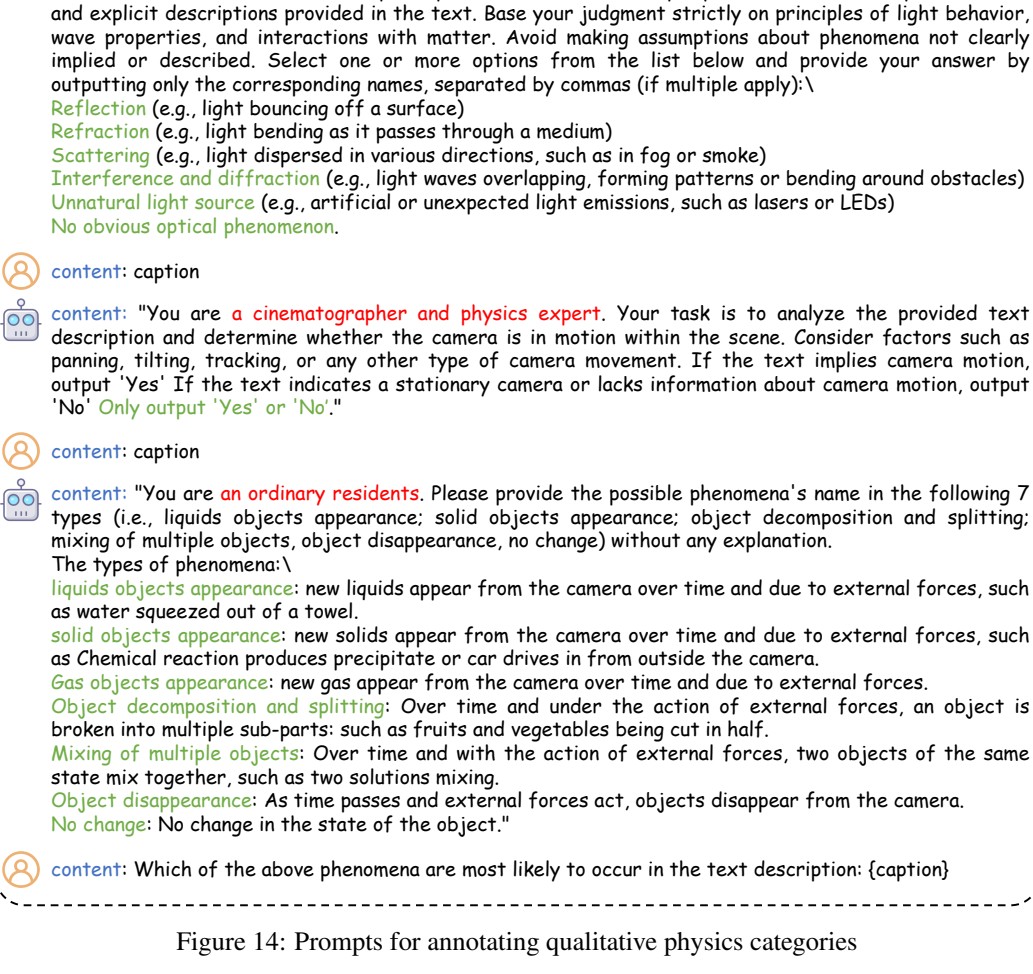

**"model": "GPT-4o-mini"**     **"role": "system"**     **"role": "user"**

content: You are a physicist with expertise in optics. Your task is to analyze the following text carefully and determine which of the listed optical phenomena are most likely represented based solely on the clear and explicit descriptions provided in the text. Base your judgment strictly on principles of light behavior, wave properties, and interactions with matter. Avoid making assumptions about phenomena not clearly implied or described. Select one or more options from the list below and provide your answer by outputting only the corresponding names, separated by commas (if multiple apply):\
Reflection (e.g., light bouncing off a surface)
Refraction (e.g., light bending as it passes through a medium)
Scattering (e.g., light dispersed in various directions, such as in fog or smoke)
Interference and diffraction (e.g., light waves overlapping, forming patterns or bending around obstacles)
Unnatural light source (e.g., artificial or unexpected light emissions, such as lasers or LEDs)
No obvious optical phenomenon.

content: caption

content: "You are a cinematographer and physics expert. Your task is to analyze the provided text description and determine whether the camera is in motion within the scene. Consider factors such as panning, tilting, tracking, or any other type of camera movement. If the text implies camera motion, output 'Yes' If the text indicates a stationary camera or lacks information about camera motion, output 'No' Only output 'Yes' or 'No'."

content: caption

content: "You are an ordinary residents. Please provide the possible phenomena's name in the following 7 types (i.e., liquids objects appearance; solid objects appearance; object decomposition and splitting; mixing of multiple objects, object disappearance, no change) without any explanation.
The types of phenomena:\
liquids objects appearance: new liquids appear from the camera over time and due to external forces, such as water squeezed out of a towel.
solid objects appearance: new solids appear from the camera over time and due to external forces, such as Chemical reaction produces precipitate or car drives in from outside the camera.
Gas objects appearance: new gas appear from the camera over time and due to external forces.
Object decomposition and splitting: Over time and under the action of external forces, an object is broken into multiple sub-parts: such as fruits and vegetables being cut in half.
Mixing of multiple objects: Over time and with the action of external forces, two objects of the same state mix together, such as two solutions mixing.
Object disappearance: As time passes and external forces act, objects disappear from the camera.
No change: No change in the state of the object."

content: Which of the above phenomena are most likely to occur in the text description: {caption}

Figure 14: Prompts for annotating qualitative physics categories

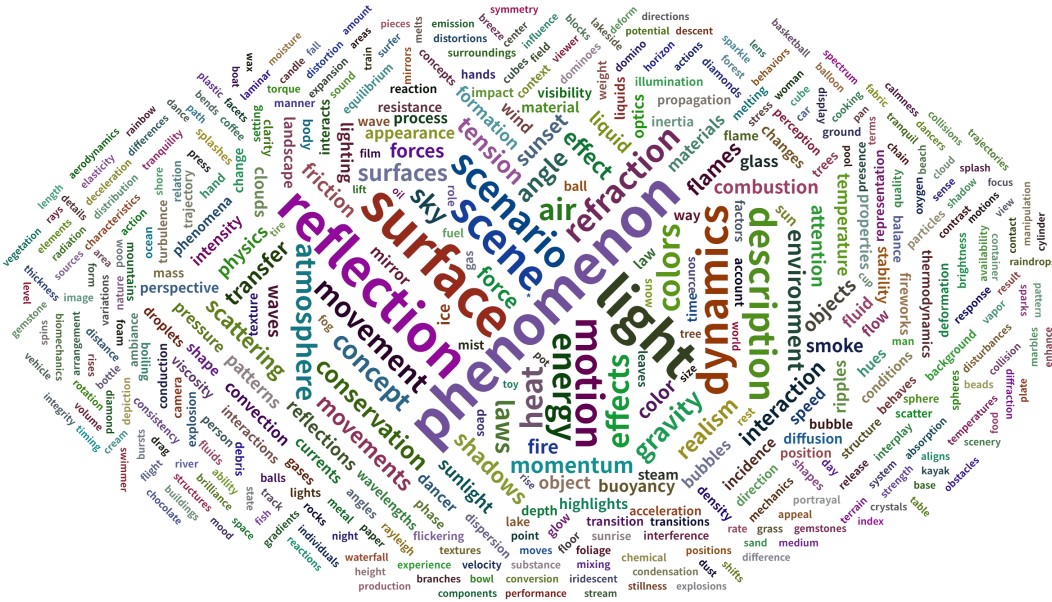

Figure 15: Word cloud generated from textual physical description, where larger words indicate higher frequencies in the dataset text

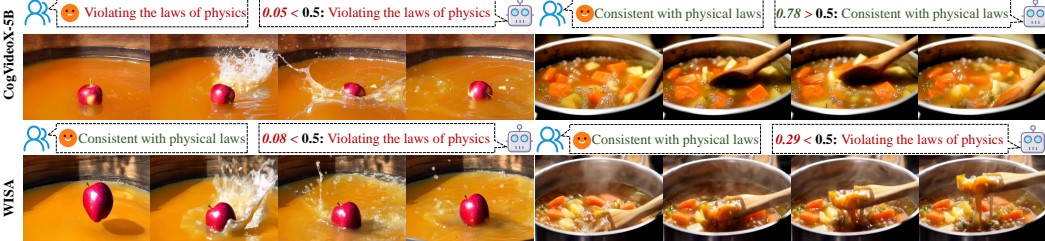

Figure 16: Human and machine evaluation results do not fully align.

