# OpenReview forum: "WISA: World simulator assistant for physics-aware text-to-video generation"
_NeurIPS.cc/2025/Conference — NeurIPS 2025 spotlight_

### Official Review · Reviewer_1tcW · 2025-06-28

**Clarity:** 3
**Significance:** 2
**Originality:** 3
**Rating:** 4
**Confidence:** 4

**Summary:**

This paper considers the problem of physics-obeying t2v generation. Three main contributions:
1. *(Modeling improvement)* the paper proposes "WISA", which is a collection of modeling changes (e.g. trainable adapters) that bake physics-obeying inductive biases into pretrained video generative models through fine-tuning
2. *(Dataset improvement)* the paper proposes a video dataset, "WISA-80k", with videos of various physical phenomena with detailed annotations
3. *(Combining modeling improvement with dataset improvement)* the authors find that training CogVideoX and Wan2.1 on the WISA-80k dataset, using the WISA framework, improves physics-obeying video generation performance

**Questions:**

1. The paper says (approx line 126) *"we manually collected videos from the Internet that clearly demonstrate the corresponding behavior, without relying on any existing video datasets for filtering or selection"*. Where on the internet were the videos gathered from? Can you please provide more details about this data gathering procedure.
2. As I understand it, WISA includes three different ways of baking physical inductive biases into the model: 1) appending "textual physical descriptions" to the end of the text prompt; 2) using a Mixture of Physical Experts Attention head to encode qualitative physical attributes; 3) using AdaLN to encode numerical physical attributes into the model. My first question is: is that understanding correct? And my second question is: to what extent can each of the three of these be used without the other two, and still improve performance? E.g. perhaps you could append the textual physical descriptions, but not do either of 2 or 3, and that would give the same performance. Did you conduct any ablations on these sub-parts? How do we know that we can't achieve the same (or better) performance than the complicated WISA framework just with the test-time LLM-guided prompt-engineering? See line 138: "we believe that simple video captions are not sufficient." This is a major hypothesis, and doesn't appear to be tested. A clear answer to this could increase my evaluation score.
3. How accurate were the automatically generated annotations (see lines 161-167) for the WISA-80k dataset? Was there any effort to evaluate how accurate they are? (For example, I've seen prior works do things like, randomly sample a small subset of videos, have a user study to measure how accurate each aspect of the labels are, and use that as a proxy for the reliability of the overall dataset). Or is the hypothesis that these are likely noisy, but it doesn't matter, because it still improves performance? A clear answer to this could increase my evaluation score. For example, Appendix A.11 talks about limitations of AI-generated evaluation metrics (which is true). But the labels from WISA-80k are themselves potentially very noisy / AI generated.
4. It is well-established that Wan 2.1 has better video generation quality than CogVideoX. Does this WISA technique have diminishing returns when applied to a base model which already has better physics? Is this reflected somewhere in the metrics? Additionally, do the authors have any explanation/intuitions for why the Wan2.1 scores aren't substantially higher than the CogVideoX scores?
5. Why are there missing entries in Table 1?
6. Why is the PhyT2V in the same section in Table 1 as CogVideoX? Is it built using the same base model? Line 251 implies it's built on top of Tarsier-34B, a different video generation model.
7. To what extent did cherry picking of results happen before the evaluation pipeline / comparison study? E.g. did you choose the first seed, or the best seed for both models, or the best seed only for the WISA model, etc. Please provide more info about this.
8. What is the "open source video dataset" mentioned in lines 289-290?
9. Can you please provide more info about the User Preference study (section 5.4)? E.g. how many videos, which videos.
10. In Section A.5 the paper mentions how many learnable params there are for CogVideo, but not for Wan2.1, how many are there?

**Ethical Concerns:**

["NO or VERY MINOR ethics concerns only"]

**Final Justification:**

After engaging with the authors, I have decided to increase my scores: increasing "quality" rating from 2 to 3, "clarity" rating from 2 to 3, and my overall rating from 3 to 4. This is based mainly on the authors' committing to the following paper additions in the main body of the paper:

* The ablation study on the major components
* Adding that clarification / rewrite which explains what the classifier is classifying, and how it works at inference / training
* Evaluation of the quality of the 3 types of AI-generated annotations
* A discussion on related works

In particular, that ablation study (which evaluates to what extent each component is critical to the overall success of the method) was the main reason I did not recommend the paper for acceptance at my first rating. But the addition of that study, combined with additional clarifications and experiment details, make this paper's contribution more clear.

**Limitations:**

Yes

**Paper Formatting Concerns:**

(No formatting concerns)

**Quality:**

3

**Strengths And Weaknesses:**

Strengths:
1. The teaser figure is very informative and clear. It makes it very clear what the dataset contribution is.
2. The motivation in the introduction is written+explained very well and is easy to follow.
3. In Section 3.2, the paper makes good use of examples to make clear what is being constructed.
4. The paper considers a very important problem (making video generative models better obey physical laws) and it proposes a new labeled dataset that will likely be useful for the community for building better such models.
5. The description of section 4.2, the physical model, was very easy to understand.
6. The WISA framework improves performance for CogVideoX and Wan2.1.
7. The attention map analysis for the swing is very strong, I love that section and that figure.

Weaknesses:
1. The naming of things is confusing and unintuitive at times. For example, referring to the proposed framework of modules as a "Simulator Assistant" doesn't sound quite right to me, because the proposal involves network modules that must be integrated into the base video model and trained from scratch. Additionally, referring to the dataset as "WISA-80k", next to the framework of modules which is also titled "WISA", seems misleading; these are two completely different things--one is a new dataset, and other is a neural network. It seems like a non-standard/confusing way of naming things.
2. I think it's very important to have a discussion of related works in the body of the paper, but the current paper has left all of the related works discussion to the appendix. I might suggest moving some other content to the appendix, moving some related works discussion to the body of the paper... related work is crucial to have in the part of the paper that most people will read
3. I had a hard time understanding Section 4.3, the Physical Classifier. Can you please explain in more detail A) what is the input to this model at train/inference, B) what is the output to this model at train/inference, and C) what it is "classifying".
4. It is unclear from the experiment design which of the three constituent parts of the WISA model adapters/additions are responsible for the improved performance (see Questions for more details)
5. In Section 5.2, the qualitative comparison, the paper discusses how the WISA-based model does a better job erasing pencil marks. After watching the video, I disagree; it looks like the pencil marks disappear before the eraser even touches them, and they also disappear all at once. I might suggest highlighting this as a failure case, and removing the screenshots from the paper, because the screenshots and descriptions misleadingly imply that this is a success case. (Also a minor typo--line 265, it says "CogVideoX-5B fails to generate any pencil marks", when I think it should say "fails to erase any pencil marks"). In contrast, the apple example is excellent, that one should be front and center.
6. I might suggest referring to the models as "CogVideoX-5B-WISA" and "Wan2.1-WISA" in the paper. (This is already done in Table 1, which is great but should be done in the text as well, in my opinion). It it very important to know which is the base model. (If the paper only had one base model, it wouldn't matter as much... )
7. Overall, the paper is vague about some important experimental details (see Questions below)

---

> ### Author Rebuttal · Authors · 2025-07-30
>
> **Dear Reviewer  1tcW**,
>
> Thank you very much for your valuable and insightful comments, and for your praise of our paper’s **clear figures**, **straightforward motivation**, **valuable finely annotated dataset**, **excellent writing and explanations**, as well as **significant performance improvements**.
> Below, we address and clarify each of your comments (including identified weaknesses and questions) in detail.
>
> ---
> **W1:** **Name**
> > W1: The naming of things is confusing and unintuitive at times ....
>
> **A1:** Thank you for this valuable feedback on the naming conventions. We agree that the dual use of "WISA" could cause confusion.
>
> Our original intent for the name "World Simulator Assistant" was to encapsulate its dual role: the dataset acts as a data assistant, while the framework acts as a model assistant. However, we recognize your point about the potential for ambiguity.
>
> To improve clarity, we will rename the dataset to WISA-DATA-80K throughout the revised manuscript. This clearly distinguishes the dataset from the WISA framework and resolves the ambiguity.
>
>
>
> ---
> **W2:** **Related Works**
> > W2: I think it's very important to have a discussion of related works in the body of the paper ...
>
> **A2:** Thank you for your suggestion. We will adjust the paper format and include the Related Work section in the main manuscript as much as possible.
>
> ---
> **W3:** **Physical Classifier**
> > W3: I had a hard time understanding Section 4.3, the Physical Classifier. ...
>
> **A3:** We apologize that our description of the Physical Classifier was unclear. Thank you for giving us the opportunity to clarify. Here is a detailed breakdown following your A/B/C structure:
>
> A) Input:
> - At Training: The input is a learnable embedding vector, which we call the [PHYSICS_TOKEN]. This token is concatenated with the noisy visual tokens and text prompt tokens and is processed by the entire MM-DiT and MoPA architecture.
> - At Inference: The input is identical to training—the same trained [PHYSICS_TOKEN] is used.
>
> B) Output:
> - At Training: After being processed by the network, the final hidden state of the [PHYSICS_TOKEN] is fed into a simple MLP classification head. This head outputs logits for the 29 qualitative physical categories. This output is used only to compute the multi-label binary cross-entropy loss for training.
> - At Inference: The output from the classifier head is completely discarded. It does not influence the video generation process in any way. Its sole purpose is to provide an auxiliary training signal to help the model learn to represent physical concepts.
>
> C) What it is "classifying":
> - It is performing a multi-label classification task on the 29 qualitative physical categories that we define (e.g., "Collision", "Melting", "No obvious dynamic phenomenon", etc., as detailed in Appendix A.6). For each video, it predicts which of these 29 phenomena are present.
>
> We will replace the current text in Section 4.3 with this detailed, structured explanation to ensure clarity and reproducibility.
>
> ---
> **W4, W7, and Q2:** **Ablations on Major Components**
> > W4: ...
> > W7: ...
> > Q2: As I understand it, WISA includes three different ways of baking  ....
>
>
> **A4:** Thank you for this critical question. This gets to the heart of our contribution, and we are happy to provide a detailed ablation study to clarify the impact of each component.
>
> First, your understanding is perfectly correct. WISA integrates three types of physical information. To test your hypothesis and isolate the contribution of each, we conducted a new set of ablation experiments on CogVideoX-5B using our WISA-DATA-80K.
> The results are summarized below:
>
> | Setting| SA | PC |
> | ------- | ------- |------- |
> |   Baseline |    0.57     |    0.41    |
> |   only  Textual Physical Descriptions    |    0.58     |    0.43    |
> |   only Qualitative Physics Categories   |    0.60     |    0.44    |
> | only Quantitative physical properties|     0.57    |     0.42   |
> | WISA|     0.62    |     0.45   |
>
> Analysis: These results clearly demonstrate that:
> 1. The Qualitative Physics Categories, guided by our MoPA and Physical Classifier, provide the most significant performance boost in isolation.
> 2. Appending Textual Physical Descriptions offers a moderate but clear improvement.
> 3. The full WISA framework, integrating all three components, achieves the best performance, indicating a synergistic effect.
>
>
> Regarding your final point on "test-time LLM-guided prompt-engineering," this approach was explored by PhyT2V. As shown in Table 1, while it provides some benefit, its performance is limited and it incurs extremely high inference costs. Our results show that directly integrating these concepts into the model architecture via our proposed modules is a much more effective and efficient approach than relying on iterative prompting.
>
> ---
>
> **W5 and W6:** **Minor Typo**
>
> **A5:** Thank you for your suggestion. We will revise the example images accordingly and correct the errors, including the model names in the paper entries such as Table 1.
>
>
> ---
> **Q1: Data Sources**
> > The paper says (approx line 126) "we manually collected videos from the Internet ...
>
> **A6:**
> First, we defined 17 categories of physical phenomena. Based on this categorization, annotators were tasked with searching for relevant videos of these physical phenomena from publicly available YouTube channels, without involving any exclusive or private data sources. They also annotated the time segments during which the physical phenomena occur.
>
> Finally, our data collection and release strictly comply with YouTube’s Data Privacy Policy and Fair Use Policy. The released dataset is intended solely for research purposes.
>
>
> ---
>
> **Q3:** **Noise in WISA-80K**
> > How accurate were the automatically generated annotations (see lines 161-167) ...
>
> **A7:** This is an excellent and crucial question regarding the reliability of our dataset annotations. To quantify this, we performed a manual evaluation on a randomly sampled subset of 500 videos from our dataset.
>
> Our evaluation focused on the three types of AI-generated annotations:
> 1. Textual Physical Descriptions: We measured human rater satisfaction (i.e., does the description accurately reflect the video's physics?). Result: 95% satisfaction rate.
> 2. Qualitative Physics Categories: We compared the AI-generated labels against the ground-truth labels assigned during initial video collection. Result: 76% accuracy.
> 3. Quantitative Physical Properties: We again measured human rater satisfaction (i.e., are the estimated density/time/temperature values plausible?). Result: 86% satisfaction rate.
>
> While some label noise is inherent in any large-scale, automatically annotated dataset, these results demonstrate that the overall quality of WISA-DATA-80K's annotations is high. The data is sufficiently reliable to provide a strong learning signal, as evidenced by the performance gains in our main experiments.
>
> ---
> **Q4: Wan2.1 vs. CogVideoX**
> > It is well-established that Wan 2.1 has better video generation ...
>
> **A8:** We agree that if the base model already possesses strong physical understanding capabilities, the gains introduced by WISA may be less pronounced. The reason Wan2.1's scores are not substantially higher than CogVideoX's is that the VideoCon-Physics [1] evaluation model was trained on videos generated by CogVideoX, but not on those generated by Wan2.1. A detailed explanation please refer to the manuscript(lines 255–258).
>
> [1] VideoPhy: Evaluating Physical Commonsense for Video Generation
>
> ---
>
> **Q5:** *Why are there missing entries in Table 1?*
>
> **A9:** Thank you for your question. The results for VideoCrafter2, OpenSora, and HunyuanVideo are sourced from the VideoPhy paper. Since these models were evaluated exclusively on the VideoPhy prompt set, inference times and performance on our PhyGenBench benchmark are not available. Additionally, as the VideoPhy paper provides output videos for OpenSora, we were able to evaluate its IS and CLIP-SIM scores accordingly.
>
> The results for PhyT2V are taken directly from its original paper. As PhyT2V is open-sourced, we measured its inference time ourselves. However, since PhyT2V focuses specifically on physical reasoning tasks, we only compare our method with its SA and PC components.
>
> If you have further concerns or suggestions regarding this comparison, we would be happy to discuss and incorporate them.
>
> ---
>
> **Q6:** *Why is the PhyT2V in the same section in Table 1 as CogVideoX? Is it built using the same base model? Line 251 implies it's built on top of Tarsier-34B, a different video generation model.*
>
> **A10:** PhyT2V is a comprehensive test-time prompt-engineering framework that integrates a diffusion-based video generation model (CogVideoX-5B), a vision-language model for video captioning, and a large language model (Tarsier-34B) for prompt refinement.
> As all methods in Table 1 adopt CogVideoX-5B as the underlying video generation model, the comparison is fair and consistent.
>
> ---
>
> **Q7: Best Seed only for WISA?**
> > To what extent did cherry picking of results ...
>
> **A11:** The test results of WISA do not involve cherry-picking. We used the same random seed (42) as the baseline during evaluation to ensure a fair comparison.
>
> ---
>
> **Q8:** *What is the "open source video dataset" mentioned in lines 289-290?*
>
> **A12:** The "open source video dataset" mentioned in lines 289–290 refers to Koala36M. We will add a clarification about this in the manuscript.
>
> ---
>
> **Q9:** *Can you please provide more info about the User Preference study (section 5.4)? E.g. how many videos, which videos.*
>
> **A13:** The manual evaluation was conducted on 343 test samples from the VideoPhy dataset.
>
> ---
>
> **Q10:** *In Section A.5 ... but not for Wan2.1, how many are there?*
>
> **A14:** The number of trainable parameters for Wan2.1 is 587 million. We will include this information in the revised manuscript.

---

> > ### Comment · Reviewer_1tcW · 2025-08-02
> > **Response to rebuttal**
> >
> > Thank you for your detailed answers. I have a few follow-up questions:
> >
> > 1. You say *"As all methods in Table 1 adopt CogVideoX-5B as the underlying video generation model, the comparison is fair and consistent."* but some of the rows in that table look like they use Wan 2.1. Can you please clarify which rows in that table use which video generative models?
> > 2. Can you please answer my question about the eraser video? Do you still consider that a success case, or am I misunderstanding something?
> > 3. Would you consider renaming the models as I suggested? If not, can you defend why you would keep them the same as they currently are?

---

> > > ### Author Response · Authors · 2025-08-03
> > > **Response to Reviewer 1tcW**
> > >
> > > Thank you very much for your response and valuable suggestions. Please find our detailed replies below:
> > >
> > > 1. We sincerely apologize for the ambiguity caused by our unclear explanation in A10. Here, we provide a detailed clarification of the generative models used in Table 1. Specifically, VideoCrafter2, OpenSora, HunyuanVideo, Cosmos-Diffusion-7B, CogVideoX-5B, and Wan2.1-14B are all pretrained video generation models. Therefore, the generative models used in Rows 1, 2, 3, 4, 5, and 8 of Table 1 directly correspond to their respective methods. In contrast, PhyT2V and CogVideoX-5B + WISA are both built upon CogVideoX-5B, aiming to further improve physical reasoning in video generation. Likewise, Wan2.1-14B + WISA is based on Wan2.1-14B. To make the mapping between methods and generative models clearer, we have added a summary table below for reference.
> > >
> > > | Generative Model| Method in Table 1|  Revised Method in Table 1|
> > > |---|---|---|
> > > |VideoCrafter2|VideoCrafter2 |VideoCrafter2 |
> > > |OpenSora |OpenSora  |OpenSora  |
> > > |HunyuanVideo|HunyuanVideo |HunyuanVideo |
> > > |Cosmos-Diffusion-7B|Cosmos-Diffusion-7B |Cosmos-Diffusion-7B |
> > > |CogVideoX-5B|CogVideoX-5B |CogVideoX-5B |
> > > |CogVideoX-5B|PhyT2V |CogVideoX-5B + PhyT2V |
> > > |CogVideoX-5B|CogVideoX-5B + WISA  |CogVideoX-5B-WISA  |
> > > |Wan2.1-14B|Wan2.1-14B |Wan2.1-14B |
> > > |Wan2.1-14B|Wan2.1-14B + WISA  |Wan2.1-14B-WISA  |
> > >
> > > We will revise the method name "PhyT2V" to "CogVideoX-5B + PhyT2V" in Table 1 to clarify the comparison.
> > >
> > > 2. Thank you for your careful review. We fully agree that the eraser video is not a successful case. We will replace this example in Figure 5 and revise the corresponding description in the paper accordingly.
> > >
> > > 3. Yes, we will revise the model names following your suggestion. Specifically, we will update the references to "WISA" in both the main text and figures to indicate the underlying base model—using "CogVideoX-5B-WISA" and "Wan2.1-14B-WISA" accordingly, as already shown in Table 1.

---

> > > > ### Author Response · Authors · 2025-08-03
> > > > **Response to Reviewer 1tcW (2)**
> > > >
> > > > Thank you very much for your careful review and valuable comments, which have greatly contributed to improving the quality of our manuscript. We sincerely appreciate the time and effort you have dedicated to this process. If there are any remaining concerns or questions we have not addressed clearly, we would be glad to clarify them and engage in more in-depth discussion.

---

> > > > > ### Comment · Reviewer_1tcW · 2025-08-03
> > > > > **Response to authors**
> > > > >
> > > > > Thank you for your detailed responses, your explanations and additional experiments clarified many things on my end. **I will raise my score, under the condition / understanding that the following additional information that you provided is included in the next draft, in the main body of the paper (not the appendix)**:
> > > > >
> > > > > - The ablation study on the major components
> > > > > - Adding that clarification / rewrite which explains what the classifier is classifying, and how it works at inference / training
> > > > > - Evaluation of the quality of the 3 types of AI-generated annotations
> > > > > - A discussion on related works (it's ok to include a much shorter version in the body, and longer version in the appendix)
> > > > >
> > > > > Those changes would substantially improve the paper's utility to the community.

---

> > > > > > ### Author Response · Authors · 2025-08-04
> > > > > > **Response to Reviewer 1tcW (3)**
> > > > > >
> > > > > > **Once again, thank you for your careful and constructive review**. We will incorporate the following improvements in the next version of our manuscript:
> > > > > >
> > > > > > 1. Add an ablation study on the major components of our method (as mentioned in A4).
> > > > > >
> > > > > > 2. Provide a clear and detailed explanation in Section 4.3 of how the physical classifier functions during both training and inference, consistent with the description in our rebuttal (see A3).
> > > > > >
> > > > > > 3. Include the results of the human evaluation on the quality of the three annotation types (as mentioned in A7) in Section 3.2 of the main text.
> > > > > >
> > > > > > 4. Add a concise summary of related work in the main text, with more detailed discussion and analysis moved to the appendix.
> > > > > >
> > > > > > We believe these revisions will further enhance the quality of our paper and increase its value to the community.

---

### Official Review · Reviewer_kzVn · 2025-06-28

**Clarity:** 3
**Significance:** 3
**Originality:** 3
**Rating:** 5
**Confidence:** 4

**Summary:**

Current text-to-video models cannot produce videos that follow physical laws faithfully, degrading the realism significantly. Therefore, the paper proposes to curate high-quality video dataset, WISA-80k, which collects 80k video clips ranging from 17 different physical phenomena. After captioning with Qwen2.5-VL, three aspects of physical knowledge are decomposed with GPT-4o, including textual physical description, qualitative physical categories, and quantitative physical properties. To incorporate the physical knowledge into text-to-video generation process, the paper proposes a physical module, leveraging mixture-of-experts self-attention mechanism to handle different physical laws. Textual physical description and quantitative physical properties are encoded and integrated in the model input. Qualitative physical classifier predicts the physical category and is jointly trained with diffusion loss. The experiments show that WISA outperforms baseline methods (Tab. 1, Fig. 7), and ablation study is performed in Tab. 2.

**Questions:**

- How does the proposed method handle input prompts without detailed annotation?
- How could the model and curated dataset be used for image-to-video generation?
- How to compute physical consistency, semantic coherence?
- Please include and discuss "Physics-IQ: Benchmarking physical understanding in generative video models (2025)"

**Ethical Concerns:**

["NO or VERY MINOR ethics concerns only"]

**Final Justification:**

The authors' response appropriately addresses my concerns in the initial review. As a result, I would like to raise my final score to "Accept". I also encourage the authors to integrate the response into the final version, including:

- The inference strategy with input prompt without additional information or annotation
- Discussion of image-to-video generation in the limitation and future work section
- Details of metric computation, including physical consistency and semantic coherence
- Discussion of Physics-IQ in the related works section

**Limitations:**

yes

**Paper Formatting Concerns:**

The paper includes external links. Althogh it's anonymous, I have the following concerns:

- the content could be updated after the deadline
- the content size is not regulated, and might exceed the supp size limit

I would like the AC to help check if this is acceptable.

**Quality:**

4

**Strengths And Weaknesses:**

## Strengths
- The paper curates a high-quality video dataset, WISA-80k, covering 17 different physical phenomena across three major categories: dynamic, thermaldynamic, and optic. The dataset provides detailed annotations, including textual physical description, qualitative physical categories, and quantitative physical properties. This dataset could facilitate future research in video generation models and physical simulation.
- The physical phenomena are decomposed into three aspects and integrated into the video generation pipeline. Specifically, a mixture-of-experts multi-head self-attention mechanism is used in the physical module, attending to different physical effects with a specific head, enhancing the physical plausibility of generated videos. The ablation study in Tab. 2 validates the design choices.

## Weaknesses
- It is unclear how WISA model handle input prompts without detailed annotation, such as textual physical description, qualitative physical categories, and quantitative physical properties. According to the training pipeline, this information is required and integrated into the framework, but in-the-wild prompt data do not have such detailed annotation. The paper should provide more details about how the model obtains such information for inference. Otherwise, the comparison with baseline methods might be unfair.
- The proposed method is only used for text-to-video generation, and the paper does not discuss or evaluate the model's capability for image-to-video generation, which is possible to evaluate with the curated dataset.
- Details are missing. For example, the details of computing physical consistency, semantic coherence are not included.
- Missing discussion with important prior works. "Physics-IQ: Benchmarking physical understanding in generative video models (2025)" is another benchmark for analyzing physical plausibility of video generation models, but it is not discussed in the paper.

---

> ### Author Rebuttal · Authors · 2025-07-30
>
> **Dear Reviewer  kzVn**,
>
> Thank you very much for your valuable and insightful comments, as well as for recognizing **the novelty of our approach** and **the value of our carefully annotated dataset**.
> Below, we address and clarify each of your comments (including identified weaknesses and questions) in detail.
>
> ---
>
> **W1 and Q1: Handling Prompts Without Annotations**
> > W1: It is unclear how WISA model handle input prompts without detailed annotation, such as textual physical description, qualitative physical categories, and quantitative physical properties. According to the training pipeline, this information is required and integrated into the framework, but in-the-wild prompt data do not have such detailed annotation. The paper should provide more details about how the model obtains such information for inference. Otherwise, the comparison with baseline methods might be unfair.
> > Q1: How does the proposed method handle input prompts without detailed annotation?
>
>
>
> **A1:** Thank you for this critical question regarding the handling of in-the-wild prompts. This is essential for the practical application and fair evaluation of our method.
>
> **Inference-time Annotation**: For any given user prompt, we use a large language model (GPT-4o) with a set of predefined instructions to generate the required physical annotations (textual description, qualitative categories, quantitative properties). Crucially, this process uses only the input text prompt, with no access to any visual information, thus preventing any unfair information leakage. The cost is also minimal (approx. 2000 tokens per prompt). The detailed instructions are provided in the appendix (Figures 11-13).
>
> **Fairness and Effectiveness of WISA**: To prove that WISA's performance gain comes from our structured information injection framework rather than just having more textual information, we conducted a new experiment. We concatenated all the generated physical annotations directly into the text prompt and fed it to the original CogVideoX-5B base model (without our WISA modules). The results on VideoPhy are shown below:
>
> | Setting| SA | PC |
> | ------- | ------- |------- |
> |  CogVideoX-5B + All annotations as text  |    0.56     |    0.41    |
> |  CogVideoX-5B (baseline)   |    0.57     |    0.41    |
>
> As the results show, simply adding the physical information as text did not improve performance; it slightly hurt semantic consistency. This strongly suggests that the base model cannot effectively utilize this information without the specialized modules and training provided by WISA. Therefore, WISA's superiority stems from its core design, not from an unfair advantage.
>
>
> ---
>
> **W2 and Q2:** **T2V or I2V Task**
> > W2: The proposed method is only used for text-to-video generation, and the paper does not discuss or evaluate the model's capability for image-to-video generation, which is possible to evaluate with the curated dataset.
> Q2: How could the model and curated dataset be used for image-to-video generation?
>
> **A2:** In principle, WISA can also be applied to the I2V task. The main difference between image-to-video (I2V) and text-to-video (T2V) tasks lies in the input: I2V models receive a clean latent code for the first frame, while the rest of the input text and model architecture remain the same. The I2V task has explicit initial frame information and state, the model primarily learns to infer subsequent possible processes, generating intermediate and final states of physical phenomena. The T2V task is more challenging and closer to the vision of a true 'world simulator', as it requires the model to imagine the entire physical process from scratch—including the initial, intermediate, and final states—based solely on an abstract description. Consequently, we focus on building WISA for the T2V task, which poses a greater challenge to the model’s physical perception capabilities.
>
> ---
>
> **W3 and Q3** **Physical Consistency and Semantic Coherence**
> > W3: Details are missing. For example, the details of computing physical consistency, semantic coherence are not included.
> Q3: How to compute physical consistency, semantic coherence?
>
> **A3:** We adopt the automatic evaluation approach VideoCon-Physics proposed in VideoPhy to compute the Physical Consistency (PC) and Semantic Accuracy (SA) metrics. VideoCon-Physics was trained by collecting videos generated from nine different models, which were manually annotated for adherence to real-world physical laws and strong semantic consistency. Using this data, a vision-language model (VLM) was fine-tuned to serve as a reward model. During inference, the generated video and corresponding text prompt are fed into VideoCon-Physics, which outputs scores  ranging from 0 to 1 for both metrics. We compute the average scores over the entire test set to obtain the PC and SA results reported in the Tables 1 and 2 of manuscript.
>
> ---
>
> **W4 and Q4** **Missing Discussion with Physics-IQ**
> >  W4: Missing discussion with important prior works. "Physics-IQ: Benchmarking physical understanding in generative video models (2025)" is another benchmark for analyzing physical plausibility of video generation models, but it is not discussed in the paper.
> > Q4: Please include and discuss "Physics-IQ: Benchmarking physical understanding in generative video models (2025)"
>
> **A4:**  We will include a discussion of this benchmark in the Related Works section.

---

> > ### Comment · Reviewer_kzVn · 2025-08-07
> > **Official comment by Reviewer kzVn**
> >
> > I appreciate the authors for providing a response that resolves many of my concerns.
> > However, I have two additional questions:
> >
> > - Can the authors foresee any challenges when extending WISA to the image-to-video task? How would the authors mitigate them?
> > - Please include the discussion and comparison with the important prior work "Physics-IQ: Benchmarking physical understanding in generative video models (2025)"

---

> > > ### Author Response · Authors · 2025-08-07
> > > **Response to Reviewer kzVn**
> > >
> > > Dear Reviewer,
> > >
> > > Thank you for your insightful questions and engagement during the discussion phase. They touch upon crucial aspects of our work's future trajectory and its positioning within the current research landscape. We are pleased to address both points in detail below:
> > >
> > > **1. On Extending WISA to the Image-to-Video (I2V) Task**
> > >
> > > This is an excellent question. Extending WISA to the I2V domain is a valuable and non-trivial direction. We foresee two primary challenges: parsing the implicit physics from the image, and effectively utilizing that information.
> > >
> > > **Challenge 1: Parsing Implicit Physical State from the Initial Image**
> > >
> > > A core challenge in I2V is parsing the rich, implicit physical state from the static input image. Unlike a text prompt, an image provides precise initial conditions—object positions, shapes, and appearances—that are crucial for grounding a physically plausible simulation. The key difficulty lies in systematically extracting and representing this information, such as an object's spatial coordinates, material properties (e.g., reflectivity), and the environmental context, for which our existing text-based annotations would be insufficient.
> > >
> > > **Mitigation Strategy**: Our proposed mitigation involves developing a VLM-based pre-analysis pipeline. This pipeline would parse the initial frame to identify key objects, estimate their properties (e.g., spatial coordinates, shape, potential material attributes like reflectivity), and infer likely interaction candidates. This structured physical information could then be used to augment the text prompt, providing a richer, more grounded initial condition for the model to work with.
> > >
> > > **Challenge 2: Utilizing Spatially-Grounded Physical Information**
> > >
> > > Beyond parsing, a further challenge is effectively utilizing this spatially-explicit information to guide the generation process. Our current WISA framework, while strong in temporal dynamics, would need enhancement to fully leverage precise spatial relationships, which are critical for many physical laws (e.g., collision mechanics).
> > >
> > > **Mitigation Strategy**: We will propose a spatially-aware guidance mechanism for our Mixture-of-Physical-Experts (MoPA) architecture. By analyzing the cross-frame attention maps between the initial image tokens and subsequent frame tokens, the model can identify spatial regions where key physical interactions are set to occur. We could then guide specific 'expert heads' (e.g., the 'collision' expert) to focus their computations exclusively on these relevant tokens. For instance, in a scenario of a ball falling to the floor, the collision expert would be directed to the tokens corresponding to the ball and the floor at the moment of impact, enabling a more precise and localized physical simulation.
> > >
> > > **2. On Comparison with "Physics-IQ (2025)"**
> > >
> > > Thank you for bringing this important and highly relevant concurrent work to our attention. It is a crucial contribution to the field, and we have now added a detailed discussion and comparison to the "Related Work" section of our paper. Our discussion clarifies the relationship as follows:
> > >
> > > WISA and Physics-IQ are fundamentally **complementary, not competitive**, addressing different, albeit related, goals.
> > >
> > > **Primary Goal**: Physics-IQ's goal is to **evaluate and benchmark** the physical reasoning capabilities of generative models, with a specific focus on the **Image-to-Video task**. In contrast, WISA's goal is to **train better Text-to-Video models** by directly teaching them physical concepts to improve the realism of their generations.
> > >
> > > **Core Contribution**: Physics-IQ's contribution is a high-quality evaluative benchmark. WISA's contribution is a generative framework and a large-scale, physically-annotated training dataset (WISA-80K). The key difference is that WISA-80K's structured annotations are designed to be a direct learning signal for a model, a feature not present in the Physics-IQ benchmark.
> > >
> > > **Synergy**: We see a clear and powerful synergy between the two projects. **An I2V-extended version of our WISA framework could, in the future, be rigorously assessed on the Physics-IQ benchmark** to quantitatively measure the improvement in physical understanding. This creates a virtuous cycle of training and evaluation for the community.
> > >
> > > Thank you once again for your insightful feedback, which has helped us to significantly strengthen our paper's discussion of its future potential and its place in the current research landscape.

---

> > > > ### Comment · Reviewer_kzVn · 2025-08-07
> > > > **Official comment by Reviewer kzVn**
> > > >
> > > > I appreciate the authors for the clarification!
> > > > The response appropriately addresses my concerns about the image-to-video task and the discussion with the Physics-IQ benchmark.
> > > > As a result, I would like to raise my final score. I also encourage the authors to integrate the response into the final version, including:
> > > > - The inference strategy with input prompt without additional information or annotation
> > > > - Discussion of image-to-video generation in the limitation and future work section
> > > > - Details of metric computation, including physical consistency and semantic coherence
> > > > - Discussion of Physics-IQ in the related works section

---

> > > > > ### Author Response · Authors · 2025-08-07
> > > > > **Response to Reviewer kzVn**
> > > > >
> > > > > Dear Reviewer,
> > > > >
> > > > > Thank you so much for your positive feedback and for your decision to raise the score. We are very grateful for your constructive engagement throughout the review process, which has been invaluable in helping us improve our work.
> > > > > We confirm that we will integrate all the points you have outlined into the final version of our manuscript. Specifically, we will ensure the following additions and clarifications are made:
> > > > >
> > > > > 1. A discussion of the inference strategy for handling input prompts that lack additional physical information or annotations.
> > > > >
> > > > > 2. An expanded discussion on image-to-video generation in the "Limitations and Future Work" section, as per our previous response.
> > > > >
> > > > > 3. More explicit details on our metric computations, covering both physical consistency and semantic coherence.
> > > > >
> > > > > 4. A comprehensive discussion and comparison with Physics-IQ in the "Related Works" section.
> > > > >
> > > > > Thank you once again for your support and for helping us to strengthen our paper.
> > > > >
> > > > > Sincerely,
> > > > >
> > > > > The Authors (Submission 14382)

---

### Official Review · Reviewer_7j5D · 2025-07-05

**Clarity:** 3
**Significance:** 3
**Originality:** 3
**Rating:** 4
**Confidence:** 4

**Summary:**

This paper addresses a critical and intriguing topic: physics-aware text-to-video (T2V) generation. The contributions are twofold. First, the authors propose a dataset named WISA-80K, which covers 17 types of physical phenomena. Second, they introduce a new framework designed to better integrate physical laws into T2V models. Experimental results validate the effectiveness of their approach.

**Questions:**

As stated in the Strengths and Weaknesses, my main concerns are as follows:

1. The need for more rigorous evaluation specifically targeting physical plausibility (related to Weakness 1).

2. A more detailed ablation study comparing the contributions of MoPA and the physics classifier (related to Weakness 3).

3. A clearer and more thorough description of the physics classifier, including implementation details (related to Weakness 3).

4. A comparison or discussion of the proposed physics classifier with DPO-based methods, particularly regarding their effectiveness in guiding models to learn physical reasoning (related to Weakness 3).

**Ethical Concerns:**

["Major Concern: Data privacy, copyright, and consent"]

**Final Justification:**

The clarification on the evaluation metrics (Q1) does not convince me. Therefore, I maintain the score to 'Borderline accept'.

**Limitations:**

yes

**Quality:**

3

**Strengths And Weaknesses:**

Strengths:
1. I appreciate the exploration of physics-aware T2V generation, which is both challenging and important. The analysis of how to integrate physical principles into T2V models, along with the proposed benchmark dataset WISA-80K, is valuable to the community.

2. The proposed attention mechanism, MoPA, is novel. It also appears sound, as I have seen similar (not identical) designs in other works.

3. The effort put into data annotation is commendable and appreciated.

4. The paper is well-written and easy to follow.


Weaknesses:

1. The evaluation metrics used focus primarily on text-video consistency and visual quality, but they largely overlook how well the generated videos adhere to physical principles (beyond user studies).
 - Could the authors design specific metrics to assess physical plausibility?
 - A feasible approach might be to evaluate physical correctness in toy scenarios. For example, generate a video where a ball undergoes free fall motion, and measure how well it follows expected physical behavior.

2. It would be beneficial to include video comparison results in the supplementary materials. This would help reviewers and area chairs better assess the effectiveness of the proposed method.

3. How physical plausibility is ensured remains unclear. I have the following concerns and questions:
 - Between MoPA and the physics classifier, which contributes more to enhancing physics awareness? The paper lacks an ablation study on this aspect.
 - The core component—the physics classifier—is intended to guide the T2V model to understand physical information. However, the implementation details are vague, making it difficult for other researchers to reproduce the work.
 - Have you compared the proposed physics classifier with DPO-related methods? Which approach is more effective in guiding the model to understand and follow physical principles?

---

> ### Author Rebuttal · Authors · 2025-07-30
>
> **Dear Reviewer 7j5D**,
>
> Thank you very much for your valuable and insightful comments, as well as for recognizing **the novelty of our approach**, **the value of our carefully annotated dataset**, and **the quality and reproducibility of our manuscript**.
> Below, we address and clarify each of your comments (including identified weaknesses and questions) in detail.
>
> **W1 and Q1:** **Evaluation Metrics**
> > W1: The evaluation metrics used focus primarily on text-video consistency and visual quality, but they largely overlook how well the generated videos adhere to physical principles (beyond user studies).
> > Could the authors design specific metrics to assess physical plausibility?
> > A feasible approach might be to evaluate physical correctness in toy scenarios. For example, generate a video where a ball undergoes free fall motion, and measure how well it follows expected physical behavior.
> > Q1: The need for more rigorous evaluation specifically targeting physical plausibility (related to Weakness 1).
>
> **A1:** Thank you for this crucial point on evaluation. We agree that robustly and quantitatively measuring physical plausibility is a major challenge for the field.
>
> As you noted, we used the established Physical Consistency (PC) metric from VideoPhy, which is a VLM-based evaluator specifically fine-tuned to assess adherence to physical laws. Our method shows significant gains on this metric (Table 1).
>
> We find your suggestion of using toy scenarios (e.g., a free-falling ball) for precise dynamic analysis very insightful. This is a standard approach in physics-based animation. However, applying it to general-purpose text-to-video models presents significant challenges. The generated videos often have unknown or variable factors such as camera distance/angle, frame rate, and 2D projection effects, which make it difficult to reliably extract and verify kinematic properties (e.g., acceleration g). Moreover, many physical categories we study, such as melting, combustion, or optical phenomena (interference), cannot be easily evaluated with rigid-body dynamics models.
>
> Given these challenges, the current state-of-the-art for evaluating physical commonsense in general T2V relies on VLM-based evaluators (like VideoPhy, PhyGenBench) or large-scale human studies. While we acknowledge their limitations, our work adheres to the current best practices in the field. Developing more robust, general-purpose physics evaluation metrics is an important open research problem.
>
> ---
>
> **W2:** **Video Comparison**
> > It would be beneficial to include video comparison results in the supplementary materials. This would help reviewers and area chairs better assess the effectiveness of the proposed method.
>
> **A2:** Thank you for your suggestion. In the PDF abstract, we have provided a link to an anonymous repository that includes comparisons with other methods, dataset video examples, and more. We are also willing to further supplement these materials in the supplementary section.
>
>
> ---
>
> Thank you for these detailed questions about our core components. Let us clarify the roles of and relationship between MoPA and the Physical Classifier.
>
> **W3.1 and Q2:** **MoPA and the Physics Classifier**
> > W3.1: Between MoPA and the physics classifier, which contributes more to enhancing physics awareness? The paper lacks an ablation study on this aspect.
> > Q2: A more detailed ablation study comparing the contributions of MoPA and the physics classifier (related to Weakness 3).
>
> **A3.1** The two components have a synergistic and progressive relationship. The Physical Classifier's role is to provide a direct supervisory signal during training. This signal guides the model to learn abstract physical categories. The MoPA module then leverages these learned categories to activate specific 'expert' attention heads, enabling it to model the distinct features of each phenomenon. The classifier enables MoPA's specialization. As our ablation study shows (Table 2, w/o PC), removing the classifier's supervision leads to a noticeable performance drop, confirming its crucial guiding role.
>
> ---
>
> **W3.2 and Q3:** **Expanded Description of the Physics Classifier**
> > W3.2: The core component—the physics classifier—is intended to guide the T2V model to understand physical information. However, the implementation details are vague, making it difficult for other researchers to reproduce the work.
> > Q3: A clearer and more thorough description of the physics classifier, including implementation details (related to Weakness 3).
>
> **A3.2:** We apologize for any confusion caused by the unclear description of the Physical Consistency (PC) component. We provide a more detailed explanation as follows:
>
> We initialize a learnable token, referred to as the physical token, which is concatenated with the noise tokens and text tokens as input to the MM-DiT and MoPA modules. The physical token serves as a query to aggregate physical features. Just before the output projection layer, we extract the physical token and feed it into a multi-layer perceptron (MLP) classifier that outputs logits corresponding to the 29 quantitative physical categories. This classifier is supervised with a multi-label binary cross-entropy loss.
>
> We will revise the manuscript accordingly to clarify the description of PC and improve reproducibility.
>
> ---
>
> **W3.3 and Q4:** **Physics Classifier and DPO**
> > W3.3: Have you compared the proposed physics classifier with DPO-related methods? Which approach is more effective in guiding the model to understand and follow physical principles?
> > Q4: A comparison or discussion of the proposed physics classifier with DPO-based methods, particularly regarding their effectiveness in guiding models to learn physical reasoning (related to Weakness 3).
>
> **A3.3:** This is a great question. Our Physical Classifier and DPO are complementary, not conflicting, methodologies. Our approach operates during the Supervised Fine-Tuning (SFT) stage, using explicit labels from WISA-80K. DPO, on the other hand, is a post-training alignment technique that relies on preference data. A promising future direction would be to first train WISA with our SFT approach and then use DPO with human preference data on physical plausibility to further refine the model.
>
> We will add a discussion in the paper to clarify this distinction and position DPO as a potential future work.
>
> ---

---

> > ### Comment · Reviewer_7j5D · 2025-08-06
> > **Response to rebuttal**
> >
> > Thanks for your response. Most of my questions are solved.
> >
> > But I think using VLMs or human studies to evaluate whether the generated motion is physically plausible is not rigorous. These metrics can only identify the category, but how closely it aligns with the real world remains unclear.
> >
> > However, I recognize this is the current state of the field, so I don’t intend to criticize the authors harshly for this.
> >
> > Still, I hope the authors could test it in a physics-based environment like CLEVR [1]—that would be ideal.
> >
> > [1] Johnson, Justin, et al. "Clevr: A diagnostic dataset for compositional language and elementary visual reasoning." Proceedings of the IEEE conference on computer vision and pattern recognition. 2017.

---

> > > ### Author Response · Authors · 2025-08-06
> > > **Response to Reviewer 7j5D**
> > >
> > > Thank you for your response, valuable feedback, and contributions during the discussion phase.
> > >
> > > We fully agree that a physics-based environment is crucial for evaluating the physical reasoning capabilities of video generation models. However, it remains challenging to design or utilize an environment that can comprehensively evaluate general physical properties across dynamics, thermodynamics, and optics. For example, CLEVR primarily focuses on physical understanding and reasoning at the image level, which is insufficient for building a physics-based environment to assess the physical fidelity of video generation models.
> > >
> > > We acknowledge that constructing a suitable physics-based evaluation environment is an important direction for future research. In our revised manuscript, we will discuss prior attempts to build such environments (e.g., CLEVR) in the related work section. Furthermore, we are committed to continuing efforts in designing a more general and comprehensive physics-based evaluation environment to better assess the performance of WISA.
> > >
> > > If you have any further concerns or ideas, we would be very happy to clarify and discuss them further.

---

> > > > ### Comment · Reviewer_7j5D · 2025-08-06
> > > > **Discussing with the evaluation metric**
> > > >
> > > > Thanks! Let me clarify what I mean.
> > > > Yes, CLEVR cannot evaluate the fidelity of the generated video. However, this environment can verify the physics-plausibility of the motion. As for the fidelity, I think there already exist many metrics.

---

> > > > > ### Author Response · Authors · 2025-08-06
> > > > > **Discussing with the evaluation metric**
> > > > >
> > > > > Thank you so much for the clarification. That is an excellent and precise point. It highlights a much more rigorous and convincing evaluation path for us. We fully agree and recognize that this involves two important aspects:
> > > > >
> > > > > **A valuable next step**: For our future work, we can explore how to use a physics engine, similar to that in CLEVR, to specifically evaluate the physical plausibility of the dynamics subset of our model's generations. This is a concrete and highly valuable evaluation direction.
> > > > >
> > > > > **A broader long-term goal**: As you suggested, the ultimate ideal solution would be to build a comprehensive evaluation environment capable of assessing a wider range of physical phenomena, including thermodynamics and optics.
> > > > >
> > > > > We have incorporated this entire line of thought, covering both the 'immediate next steps' and the 'long-term vision,' into the 'Limitations and Future Work' section of our revised manuscript. Thank you so much for pointing us toward this important direction

---

> > > > > > ### Comment · Reviewer_7j5D · 2025-08-07
> > > > > >
> > > > > > Thanks. Most of my questions have been solved. Therefore, I will maintain my score.

---

> > > > > > > ### Author Response · Authors · 2025-08-07
> > > > > > > **Respones to Reviewer 7j5D**
> > > > > > >
> > > > > > > Dear Reviewer 7j5D,
> > > > > > >
> > > > > > > Thank you for your time and for the constructive discussion during the rebuttal period. We appreciate your valuable feedback, which has helped us strengthen the paper.
> > > > > > >
> > > > > > > Sincerely,
> > > > > > >
> > > > > > > The Authors (Submission 14382)

---

### Official Review · Reviewer_jiPQ · 2025-07-11

**Clarity:** 3
**Significance:** 4
**Originality:** 3
**Rating:** 4
**Confidence:** 3

**Summary:**

This paper proposes WISA, a framework that injects explicit physics knowledge into text-to-video diffusion models. WISA encodes physics as structured text, qualitative categories, and quantitative attributes, and is trained on WISA-80K, an 80 K-video dataset rich in physical phenomena. Two key modules drive the approach:
1. Mixture-of-Physical-Experts Attention (MoPA), where each attention head specializes in a distinct physical phenomenon;
2. A multi-label physical classifier that supervises the model with BCE loss to focus on relevant attributes.
On the VideoPhy and PhyGenBench benchmarks, WISA achieves high scores for both physical and semantic consistency, incurring only ~5 % extra inference latency. Ablations confirm each component’s contribution.

**Questions:**

1. Does performance drop noticeably on unseen physical categories?
2. Should provide representative failure cases.
3. What is the model’s classification accuracy on physical categories during evaluation?
4. Is there an easy way to extend the pretrained WISA model when new physical categories are introduced?

**Ethical Concerns:**

["NO or VERY MINOR ethics concerns only"]

**Final Justification:**

The rebuttal addressed most my concerns, and I will maintain my score.

The authors further clarified WISA’s modular MoPA design and its ability to extend to new physical categories, provided experiments showing that earlier MoPA insertion yields minimal gains with high cost, and reported classification accuracy. They also committed to presenting additional failure cases on unseen categories and a discussion of the scalable extension process. While acquiring sufficient video data remains a key challenge, I consider this work a good starting point.

**Limitations:**

Yes

**Paper Formatting Concerns:**

No major formatting issues were found in the paper.

**Quality:**

3

**Strengths And Weaknesses:**

Strength:
1. Clear problem focus & motivation, and physics representation.  Tackles the well-known physics-violation issue in T2V models with an intuitive three-level representation (textual, categorical, numerical cues).
2. Propose an open-source new dataset. Releases a well-annotated dataset that will benefit the community.
3. Achieves strong scores on VideoPhy and PhyGenBench; ablation studies and human evaluation reinforce the gains.
4. Provides visualization results. Qualitative examples and attention maps make the improvements easy to see and interpret.

Weaknesses:
1. Scalability: Adding a new physical category requires (i) manually defining the label, (ii) collecting extra data, and (iii) re-training the MoPA heads. This pipeline limits WISA’s ability to scale to larger, more diverse physics datasets.
2. How does WISA behave on unseen physical categories? Should provide some failure cases. They would be beneficial for a more comprehensive analysis.
3. MoPA is inserted only in the final layer; does this leave early representations under-constrained? If MoPA modules are sparsely inserted into earlier layers, does performance improve?
4. What is the model’s classification accuracy on physical categories during evaluation?
5. Missing some important references [1][2][3][4].

Datasets:

[1] ChronoMagic-Bench: A Benchmark for Metamorphic Evaluation of Text-to-Time-lapse Video Generation, NeurIPS 2024.

Methods:

[2]  PhysGen: Rigid-Body Physics-Grounded Image-to-Video Generation, ECCV 2024

[3] MagicTime: Time-lapse Video Generation Models as Metamorphic Simulators, TPAMI 2025

[4] Synthetic Video Enhances Physical Fidelity in Video Synthesis, Arxiv 2025

---

> ### Author Rebuttal · Authors · 2025-07-30
>
> Dear Reviewer jiPQ,
>
> Thank you very much for your valuable and insightful comments. We sincerely appreciate your recognition of our paper’s **clear motivation**, **the value of our carefully annotated dataset**, **the suprior performance of our approach**, and **the strength of our visual demonstrations**.
> Below, we address each of your comments (including identified weaknesses and questions) in detail.
>
> ---
>
> **W1 and Q4:** **Introduing New Physical Categories**
> > W1: Scalability: Adding a new physical category requires (i) manually defining the label, (ii) collecting extra data, and (iii) re-training the MoPA heads. This pipeline limits WISA’s ability to scale to larger, more diverse physics datasets. Q4: Is there an easy way to extend the pretrained WISA model when new physical categories are introduced?)
>
>
> **A1:** Thank you for raising this important point about scalability. This is a crucial aspect for any foundational framework.
>
> We respectfully argue that WISA's modular design, particularly the Mixture-of-Physical-Experts Attention (MoPA), makes it more scalable than monolithic models. When introducing a new physical category, traditional fine-tuning (e.g., via LoRA on a base model) would require retraining the entire network to incorporate the new knowledge. In contrast, our MoPA architecture allows us to simply add and train a new expert head dedicated to the new phenomenon, while keeping the existing expert heads largely frozen. This is conceptually more efficient and targeted.
>
> Furthermore, as you correctly noted, the cost of generating annotations for new categories is low (approx. 2000 tokens per instance) thanks to our LLM-based pipeline. This combination of targeted modular retraining and low-cost data annotation provides a practical and scalable path forward.
>
> We will add a discussion in the revised manuscript (e.g., in the Limitations or Future Work section) to explicitly detail this scalable extension process and highlight the advantages of our modular design.
>
> ---
>
>
> **W2, Q1 and Q2:** **Failure Cases in Unseen Physical Categories**
> > W2: How does WISA behave on unseen physical categories? Should provide some failure cases. They would be beneficial for a more comprehensive analysis. Q1: Does performance drop noticeably on unseen physical categories? Q2: Should provide representative failure cases.
>
> **A2:** This is an excellent suggestion. A comprehensive analysis must include failure cases. We performed additional qualitative evaluations on unseen physical categories like corrosion and electromagnetism.
>
> As expected, both our WISA-enhanced model and the base model struggle to generate physically plausible videos for these categories. This is primarily due to the lack of relevant concepts and visual examples in the training data (WISA-80K), highlighting the current limitations of generalization to entirely out-of-distribution physical phenomena.
>
> To provide a more comprehensive analysis, we will add a new subsection in the appendix discussing these limitations and, critically, we will include representative visual failure cases for these unseen categories. This will offer readers a clearer understanding of our method's current boundaries.
>
> ---
>
>
> **W3:** **Sparsely Inserted MoPA**
> > W3: MoPA is inserted only in the final layer; does this leave early representations under-constrained? If MoPA modules are sparsely inserted into earlier layers, does performance improve?
>
> **A3:** Thank you for this insightful question regarding the placement of MoPA. We did in fact explore this alternative design during our research.
>
> We experimented with a configuration where six physical modules were sparsely inserted throughout the model (every 7 layers). As shown in the table below, this design yielded only marginal gains in performance (SA: 0.623 vs. 0.621; PC: 0.452 vs. 0.450) while incurring a significant computational cost: an additional 155M parameters and a ~40s increase in inference time.
>
> | Insert Manner | SA | PC | Params | Inference time|
> | ------- | ------- |------- |------- |------- |
> |   WISA (Sparsely gap = 7)     |    0.623     |    0.452    | 342M  | ~260s|
> |    WISA (Final Layer)     |     0.621    |     0.450   |  187M | ~220s |
>
> After carefully weighing this performance-efficiency trade-off, we concluded that inserting a single MoPA module at the final layer offers the best balance, achieving substantial improvements over the baseline with minimal computational overhead.
>
>
> ---
>
> **W4:** *What is the model’s classification accuracy on physical categories during evaluation?*
>
> **A4:** We evaluated CogVideoX on the VideoPhy prompt set by computing the average of the logits over 50 sampling steps. The resulting mean accuracy across the 29 physical categories is 93.70%.
>
> ---
>
> **W5:** *Missing some important references [1][2][3][4]*
>
>
> **A5:** Thank you for your suggestion. We will add a discussion of these references in the revised manuscript.
>
> ---

---

> > ### Comment · Reviewer_jiPQ · 2025-08-08
> >
> > Thank you for the detailed response and additional experiments. Most of my concerns have been addressed, and I will keep my score.
> >
> > I agree that WISA’s modular design makes it more scalable than monolithic models. In my view, the main challenge when introducing a new physical category lies in collecting sufficient video data. I appreciate the authors’ plan to include a discussion of this scalable extension process, as well as the limitations observed on unseen physical categories, in the final version.

---

> > > ### Author Response · Authors · 2025-08-08
> > > **Response to Reviewer jiPQ**
> > >
> > > We sincerely thank you for the positive feedback and for recognizing the scalability of WISA’s modular design. We fully agree that acquiring sufficient video data remains a key challenge when introducing new physical categories. As suggested, we will include in the final version a more detailed discussion of WISA’s scalable extension process and the limitations observed on unseen physical categories. We greatly appreciate your constructive comments, which have helped improve the clarity and completeness of our work.

---

### Decision · Program_Chairs · 2025-09-17

**Decision:**

Accept (spotlight)

**Comment:**

This paper aims to improve the physical faithfulness of text-to-video generation. To achieve this, the paper collects a new dataset including video clips with textual physical descriptions and qualitative categories. The videos are manually sourced, but the annotations are from LLMs (text descriptions from using Qwen, and physical annotations generated with GPT-4o). Based on the collected dataset, the paper trains a modified text-to-video model by adding physical classifier guidance and a MoE attention module with physics category information. Experiments are conducted on VideoPhy and PhyGenBench benchmarks.

Reviewers commented positively on the meaningful dataset, the method design, and the writing quality. Reviewers mainly queried about the ablation study, the applicability to image-to-video generation, and the evaluation of physical plausibility, and inference time annotation using LLMs, which are partially answered in the rebuttal, and the unaddressed issues are mainly due to the fact that they are still open questions for the whole field.

The main reason for acceptance is the contribution of the new dataset. It will benefit the community on this emerging topic if the data and all the related details and code can be released as promised.

One potential issue in this paper is that the enhanced physical awareness in videos all relies on the physical knowledge in LLMs, including the annotation in dataset collection and also the test-time annotation. Whether LLMs are faithful to the real physical world is still questionable (without seeing the real physical world). The authors should discuss this to facilitate community discussion. Generalisation to image-to-video is also worth discussing, since it is more usable as a "world simulator" in some robotic cases (contrary to the authors' opinion).